# Role of Radiation Therapy for Biliary Tract Cancers

**DOI:** 10.3390/curroncol32100545

**Published:** 2025-09-28

**Authors:** Molly A. Chakraborty, Ritesh Kumar, Brett L. Ecker, Haejin In, Russell C. Langan, Mariam Eskander, Salma K. Jabbour

**Affiliations:** 1Rutgers New Jersey Medical School, Newark, NJ 07103, USA; mac865@njms.rutgers.edu; 2Department of Radiation Oncology, Rutgers Cancer Institute, New Brunswick, NJ 08901, USA; rk912@cinj.rutgers.edu; 3Department of Surgery, Rutgers Cancer Institute, New Brunswick, NJ 08901, USA; brett.ecker@rutgers.edu (B.L.E.); hi80@cinj.rutgers.edu (H.I.); russell.langan@rwjbh.org (R.C.L.); me550@cinj.rutgers.edu (M.E.)

**Keywords:** radiotherapy, biliary tract cancers, cholangiocarcinoma, gallbladder cancer, ampullary cancer

## Abstract

**Simple Summary:**

Biliary tract cancers include cholangiocarcinoma (cancer of the bile ducts), gallbladder cancer, and ampullary cancer (cancers occurring at confluence of the common bile duct and pancreatic duct). Survival outcomes for biliary tract cancers are poor despite surgical intervention, and many patients are unable to undergo surgery due to advanced disease at the time of cancer diagnosis. Thus, radiotherapy, either before surgery, after surgery, or in place of surgery with either curative or palliative treatment intent, may improve outcomes for patients with biliary tract cancers. Herein, we discuss the current literature investigating radiotherapy for biliary tract cancers. Most studies show a benefit to radiotherapy, with the strongest evidence for those with advanced-stage disease. Further study is needed, particularly prospective clinical trials and evaluation of contemporary radiotherapy techniques that may allow for dose escalation (e.g., proton therapy and stereotactic body radiotherapy).

**Abstract:**

Biliary tract cancers include cholangiocarcinoma, gallbladder cancer, and ampullary cancer. Although overall rare, the incidence is increasing globally, particularly the subset of intrahepatic cholangiocarcinoma. Surgery is currently considered to be the only curative treatment approach; however, survival outcomes after surgery remain poor. Moreover, many patients already have advanced-stage, unresectable disease at the time of diagnosis. Herein, we will review the role of adjuvant radiotherapy to improve local control after surgery, the role of neoadjuvant radiotherapy to increase the proportion of patients able to undergo surgery, and the use of definitive/palliative radiotherapy to provide local control/symptom relief for patients who have inoperable disease. Most studies observed a survival benefit associated with radiotherapy, with the strongest evidence for those with high-risk disease features (e.g., positive surgical margins, lymph node involvement). However, due to the low incidence of biliary tract cancers, most existing studies are retrospective; there is very limited randomized data and prospective studies tend to have small sample sizes, underscoring the need for more high-quality research on radiotherapy for biliary tract cancers. As some studies show evidence of a dose-dependent response, further investigation into the delivery of dose-escalated radiotherapy with modern techniques such as proton therapy is warranted.

## 1. Introduction

Biliary tract cancer (BTC) refers to a spectrum of malignancies, including cholangiocarcinoma (CCA), gallbladder cancer (GBC), and ampullary cancers. CCA refers to cancer of the epithelium of the bile ducts and can be further classified into extrahepatic (EHCC) and intrahepatic (IHCC) subtypes. Within EHCC, there can be further classification into hilar/perihilar (pCCA), which is defined as occurring distal to the second-order biliary ducts, but proximal to the insertion of the cystic duct, and distal (dCCA), which is defined as originating distally from the insertion of the cystic duct [1]. Although ampullary cancers may not traditionally be considered a subtype of BTC, as these cancers are often grouped with dCCAs, we included this disease site in this review for completeness.

Although rare, BTC incidence, especially IHCC incidence, is rising globally and already poses a significant health problem in endemic areas. Surgery is currently considered the approach of choice for treatment of BTCs; however, many patients present with advanced, unresectable disease, suggesting a potential role for neoadjuvant, definitive, or palliative radiotherapy (RT). Additionally, despite surgical intervention, survival and disease control outcomes for BTCs remain poor, especially for patients with positive margins or involved lymph nodes, suggesting a potential role for adjuvant RT [1,2].

Due to the uncommon incidence of BTCs, there is a lack of randomized clinical trial data on the efficacy of RT for BTCs, and many existing retrospective or phase I/II clinical trials have small sample sizes. Additionally, many studies include multiple sites of BTCs or also include non-biliary tract malignancies, making it difficult to draw conclusions about RT efficacy in individual BTC sites. Herein, we review the existing literature on RT for the treatment of BTCs.

## 2. Etiology/Risk Factors

There is wide variation in the incidence of CCA globally. Regions in East Asia, where liver flukes, such as *Opisthorchis viverinni* and *Clonorchis sinensis*, are endemic, can report rates of CCA as much as 100-times greater than the rates observed in Western countries [1,3]. Outside of these areas, the strongest risk factor for the development of CCAs are choledochal cysts [3]. Other risk factors are contributors to chronic inflammation or cholestasis, which result in a cycle of cellular damage and repair, eventually leading to carcinogenesis [4]. Some of these risk factors include cholelithiasis, cirrhosis, Hepatitis B and C, alcohol use, primary sclerosing cholangitis, Type II diabetes mellitus, and smoking [3,4].

Similarly, chronic inflammatory states are also thought to lead to the development of GBC. The presence of gallstones is likely the most significant risk factor for the development of GBC, and greater size of gallstones has been found to be associated with an increased risk [5,6]. This suggests that risk factors for gallstones, such as increased body weight or high cholesterol levels, may also increase the risk for GBC [7]. Other risk factors include autoimmune disorders and chronic infection [6].

There has been less study on potential risk factors for ampullary cancer, but some may include cholelithiasis, components of metabolic syndrome, chronic pancreatitis, and proton pump inhibitor use [8,9].

## 3. Anatomy

IHCC refers to CCA originating from proximal to the second-order bile ducts. The lymphatic drainage and spread of IHCC depends on the location within the liver: tumors in the left liver are more likely to drain to nodes within the celiac basin while tumors in the right liver are more likely to drain to portocaval lymph nodes [10]. pCCA is a CCA that originates distally to the second-order bile ducts, but proximally to the joining of the cystic and common bile ducts. Regional lymph nodes include portal, hilar, cystic duct, choledochal, hepatic arterial, and posterior pancreaticoduodenal nodes [10].

dCCA refers to CCA originating distally to the confluence of the cystic duct and common bile duct. Muscle fibers are most prominent in the distal segment of the common bile ducts; however, extrahepatic ducts lack a serosa. Regional lymph nodes include nodes along the right side of the superior mesenteric artery, the anterior and posterior pancreaticoduodenal artery, the hepatic artery, and the common bile duct [10].

The gallbladder consists of the fundus, body, and neck, which attaches to the cystic duct. Of note, the gallbladder lacks a submucosal layer and the side of the gallbladder bordering the liver lacks a serosa. Regional lymph nodes are considered to be the nodes along the cystic duct, common bile duct, hepatic artery, and portal vein [10].

The ampulla of Vater is the opening of the pancreatic duct and common bile duct into the posteromedial wall of the duodenum through a mucosal elevation referred to as the papilla of Vater. The sphincter of Oddi surrounds the distal pancreatic duct and common bile duct. Regional lymph nodes include the nodes along the portal vein and hepatic artery and the peripancreatic lymph nodes [10].

## 4. Histology

Macroscopically, CCAs tend to grow in three main subtypes: mass-forming, periductal-infiltrating, and intraductal-growth. The intraductal-growth subtype tends to have the best prognosis, as it is more likely to be diagnosed at an early stage compared to the other subtypes. Most CCAs are adenocarcinomas, but there are other rare microscopic patterns [11].

IHCC is commonly divided into large and small duct subtypes. IHCC of the large-duct subtype is derived from cholangiocytes of the hepatic ducts, which have similar histological features to the epithelia of the pancreatic duct. These cells tend to be tall, cylindrical, and produce mucin. IHCC of the small-duct subtype is derived from cholangiocytes of the small bile ducts (ductules), which tend to be cuboidal and not produce mucin [11,12]. The histologic appearance of pCCA and dCCA tend to be similar to the large-duct subtype of IHCC [11].

Most GBCs are derived from the fundus. Adenocarcinoma is the most common microscopic subtype, of which there are several sub-patterns. Biliary-type adenocarcinoma is the most common of these, but others include intestinal-type and mucinous adenocarcinoma. Non-adenocarcinoma subtypes include adenosquamous, squamous cell, hepatoid, and sarcomatoid carcinomas [13].

Ampullary cancers are most commonly divided into intestinal and pancreaticobiliary subtypes. The intestinal subtype has morphologic features similar to colonic adenocarcinomas. The pancreaticobiliary subtype is more common, and has morphologic features typical of pancreatic–biliary carcinomas. Some ampullary tumors may have mixed features of both subtypes. Some studies have suggested that pancreaticobiliary ampullary cancers are more likely to present with high-risk features and result in worse survival; however, other studies did not report any differences between the subtypes [14].

## 5. Radiotherapy for Cholangiocarcinoma

CCA is rare and accounts for only about 3% of gastrointestinal malignancies. CCA has a poor prognosis; even with surgical resection, the 5-year overall survival (OS) is estimated to be about 20–40% depending on the site of the tumor. About 10–30% of patients have resectable disease at presentation [15]. Given these data, there is potential for RT to be beneficial as a part of multimodal treatment. Positive margins and lymph node involvement are associated with worse survival following the surgical resection of CCAs, suggesting a potential role of adjuvant RT [1,15,16]. Surgery is currently the primarily accepted curative treatment approach, thus tumor downstaging via neoadjuvant RT may be beneficial for patients with inoperable disease. For palliative treatment, systemic therapy is the backbone for management for CCAs. However, patients with advanced CCA have a high risk of developing biliary obstruction, which can lead to subsequent infection/sepsis that can often be life threatening. Palliative RT may help to prevent or relieve obstructive symptoms and may be more tolerable than CT [1].

### 5.1. Intrahepatic Cholangiocarcinoma

IHCC is considered a form of primary liver cancer and is the second most common type after hepatocellular carcinoma (HCC). IHCC originates from the epithelia of the bile ducts within the liver, whereas HCC originates from hepatocytes. There were an estimated 806,000 new cases of primary liver cancer globally in 2020, with IHCC comprising about 10–15% of these cases [17]. Although rare in developed countries, the incidence of IHCC has been rising. In the United States, the incidence increased from 0.44 to 1.18 cases per 100,000 people between 1973 and 2012 [18].

#### 5.1.1. Adjuvant Radiotherapy

According to the National Comprehensive Cancer Network (NCCN) guidelines, if a patient can undergo upfront surgery, this is the preferred initial approach to treatment. Some contraindications to surgical management include multifocal disease, nodal involvement distant to the porta hepatis, and metastatic disease [19]. For patients who can undergo upfront surgery, the most common approach to treatment is segmental or major hepatic resection with portal lymphadenectomy [1,15,19]. After surgery, 5-year survival is approximately 25–40%, and high-risk disease features such as positive lymph nodes and non-R0 resection have been associated with worse outcomes [1]. Given low survival after surgery, several studies have investigated the use of adjuvant RT for patients with resectable IHCC, although to our knowledge, there are a lack of prospective data and retrospective data are conflicting.

Lin et al. (Table 1) conducted a study of 599 patients with resectable IHCC from the Taiwan Cancer Registry Database. A total of 320 patients received RT, either with concurrent or sequential CT. The median RT dose was 50.4 Gy and patients who received a dose of less than 45 Gy were excluded. In the concurrent CT and RT (CRT) group, 64.9% of patients had stage III+ disease, while 54.1% and 49.4% had stage III+ disease in the sequential CT followed by RT (CT+RT) and no adjuvant therapy groups, respectively. A total of 74.7% of patients in the CRT group had positive surgical margins, while 66.4% and 59.9% of patients had positive margins in the CT+RT and no adjuvant therapy groups, respectively. Despite the greater proportion of advanced disease or positive margins in the CRT group, the 2-year OS was higher in the CRT group than the CT+RT and no adjuvant therapy groups, although the difference was not significant (47.8%, 33.4%, and 38.0%, respectively, *p* = 0.113). CRT was found to be associated with better OS in patients with advanced disease (stages III and IV) compared to no adjuvant therapy (HR: 0.55, 95% CI 0.41–0.74) and was also associated with better OS for patients with early-stage disease (stages I and II) with positive margins (HR: 0.65, 95% CI 0.56–0.92). CT+RT was not associated with better OS than no adjuvant therapy. No direct comparison between CRT and CT+RT was performed. The results of the Cox regression demonstrate that similar to other BTC sites, evidence for the efficacy of adjuvant RT is strongest for patients with high-risk disease features, such as positive surgical margins or lymph node involvement [20].

Tran Cao et al. (Table 1) conducted a study of 2344 patients from the National Cancer Database (NCDB) with T1-T3N1M0 GBC or IHCC. Of these patients who underwent surgery for resectable IHCC, 79 received adjuvant CRT, 74 received adjuvant CT only, and 170 patients received no adjuvant therapy. A total of 48.1% of patients in the CRT group, 48.7% of patients in the CT group, and 38.2% of patients in the no adjuvant therapy groups had T3 disease, respectively. A total of 39.3% in the CRT group, 28.5% in the CT group, and 32.4% in the no adjuvant therapy group had positive surgical margins. The median OS was 22.6 months for patients receiving adjuvant therapy versus 16.2 months for patients receiving surgery only. The study concluded that adjuvant CRT did not provide a survival benefit for patients with IHCC with negative surgical margins (HR: 0.84, 95% CI 0.52–1.32) or with positive surgical margins (HR: 0.51, 95% CI 0.48–1.03) compared to surgery alone [21]. However, the results must be interpreted with caution as patients in the CRT group were more likely to have factors associated with worse prognosis such as a higher tumor stage and positive margins in this nonrandomized comparison.

Overall, there is currently limited evidence for adjuvant RT for IHCC. Existing data suggest that adjuvant CRT is more likely to be of benefit compared to RT alone or CT+RT, and is most likely to improve outcomes for patients with high-risk disease features such as positive surgical margins or lymph node involvement (Table 1). The current NCCN guidelines recommend systemic therapy as the preferred approach to adjuvant treatment, but list adjuvant CRT or CT combined with CRT as options to consider if the patient has R1 margins or positive regional nodes [19].

#### 5.1.2. Neoadjuvant Radiotherapy

As only approximately 21.9% of IHCC cases are resectable, neoadjuvant RT could be beneficial for tumor downstaging, allowing for subsequent surgery [22]. Cho et al. (Table 1) performed a retrospective study including 64 patients receiving neoadjuvant CRT or CT+RT (CT consisted of 5-fluorouracil (5-FU), gemcitabine, or gemcitabine with cisplatin). The clinical target volume (CTV) was defined as the gross tumor volume (GTV) plus a 5 mm margin, but was manually modified to reduce the dose to the duodenum, stomach, and chest wall. The planning target volume (PTV) was an additional 5–10 mm margin depending on the observed diaphragmatic movement during respiration. A total of 12.5% of patients were able to undergo curative surgery with negative margins following neoadjuvant therapy. A quarter of patients showed a partial radiological response after neoadjuvant therapy. The 3-year locoregional free survival (LRFS) was 50% and 15.7% for patients who were able to undergo surgery versus those who were not, respectively (*p* = 0.03). The 3-year OS was 50% and 11.2% in these same groups, respectively (*p* = 0.012). This study found that a higher RT dose (a biological effective dose (BED) > 55 Gy) and the delivery of gemcitabine-based CT was associated with higher rates of curative surgery [23]. These data suggest that only a few patients are able to undergo curative surgery after neoadjuvant CRT or CT+RT, but those who are able to undergo surgery may gain a significant survival benefit. Additionally, dose escalation may allow for a greater proportion of patients to undergo curative surgery.

Sumiyoshi et al. (Table 1) performed a small study including 15 patients with either unresectable pCCA (eight) or IHCC (seven). The patients were all treated with neoadjuvant CRT. The majority of patients (93.3%) received the S-1 CT regimen. All patients received intensity-modulated RT (IMRT) to a total dose of 50 Gy in 25 daily fractions (fx). The GTV was defined as the area of solid macroscopic tumor, and for patients with broad extra-hepatic perineural invasion (PNI), this also included the area around the proper hepatic and common hepatic arteries. Of patients with IHCC, 71.4% were able to undergo surgery and 85.7% showed evidence of tumor shrinkage. For all included patients able to undergo surgery, the median survival was 37 months, compared to 10 months for patients unable to undergo surgery [24]. This study resulted in a greater proportion of patients compared to Cho et al. who were able to eventually undergo surgery, but this should be interpreted with caution due to the small sample size. Similar to Cho et. al, patients who were able to undergo surgery gained a large survival benefit [23,24].

There is overall limited existing evidence on neoadjuvant RT for patients with IHCC. Studies show varying rates of patients able to undergo surgery following neoadjuvant RT, but demonstrate that those who are able to undergo surgery likely benefit substantially and dose escalation may allow for more patients to undergo surgery (Table 1). The NCCN guidelines suggest CT followed by CRT or CRT alone as potential options for patients with unresectable IHCC, and also recommend re-evaluation for surgical treatment following initial therapy [19].

**Table 1 curroncol-32-00545-t001:** Selected studies of operative management of IHCC that include RT.

Study	Type	*n*	Population	Treatment	Key Results
Shinohara et al. (2008) [25]	Database Retrospective	3839 total; 7% received adjuvant RT	Patients with IHCC	Surgery alone, RT alone, Surgery + adjuvant RT, or no treatment	Median survival was 11 months in surgery + adjuvant RT group, 6 months in surgery alone group, 7 months in RT alone group, and 3 months for no treatment group Surgery + adjuvant RT led to significantly better survival than surgery only (*p* = 0.014)
Nantajit et al. (2016) [26]	Single-institution Retrospective	27 total; 14 had IHCC	Patients with resectable CCA	+/− Surgery → CRT or RT and CT (mean RT dose 48.3 Gy)	2-year OS: 40.7% 2-year OS in patients receiving > 54 Gy: 57.1% 2-year OS in patients receiving < 54 Gy: 23.1%
Cho et al. (2017) [23]	Single-institution Retrospective	120	Patients with IHCC	CRT → +/− Surgery	12.5% of patients were able to undergo curative surgery after neoadjuvant therapy 3-year LRFS was 50% vs. 15.7% in surgery and non-surgery groups, respectively 3-year OS was 50% vs. 11.2% in surgery and non-surgery groups, respectively Higher RT dose (>55 Gy) associated with higher rate of curative surgery
Lin et al. (2018) [20]	Database Retrospective	599 total; 320 received RT	Patients with resectable IHCC	Surgery → CRT or CT+RT	CRT was associated with better survival compared to no adjuvant therapy in patients with advanced disease (HR: 0.55, 95% CI 0.41–0.74), or patients with early disease with positive margins (HR: 0.65, 95% CI 0.56–0.92)%
Tran Cao et al. (2018) [21]	Database Retrospective	2344; 79 received adjuvant CRT for IHCC	T1-T3N1M0 GBC or IHCC	Nonoperative vs. surgery vs. surgery + adjuvant therapy (CRT or CT only)	CRT did not provide survival benefit for patients with IHCC regardless of margin status: negative surgical margins (HR: 0.84, 95% CI 0.53–1.3), positive surgical margins (HR: 0.51, 95% CI 0.48–1.03).
Sumiyoshi et al. (2018) [24]	Single-institution Retrospective	15 total; 7 with IHCC	Patients with locally advanced unresectable IHCC or pCCA	CRT (RT dose 50 Gy) → +/− Surgery	Of 7 patients with IHCC, 5 became candidates for surgery and 6 showed tumor shrinkage The median survival time of all patients able to undergo surgery was 37 months, compared to 10 months for those were unable to undergo surgery
Mukai et al. (2019) [27]	Single-institution Retrospective	32	Patients with resectable CCA	Surgery (partial liver resection) → RT (median dose 50 Gy)	2-year OS: 72.4% 2-year disease free survival (DFS): 47.7% Median OS: 40 months
Wong et al. (2019) [28]	Prospective	18	Patients with locally advanced hilar CCA or IHCC planning to undergo liver transplantation	SBRT (40 Gy/5 fx) → CT → Liver transplantation	11 patients did not complete neoadjuvant protocol due to progression or severe toxicity 1-year post transplant survival: 80% Median OS in dropout patients: 14.4 months
Rodriguez et al. (2022) [29]	Database Retrospective	875 total; 57 received CRT	Patients with resectable CCA	Surgery → CT or CRT (median RT dose 50 Gy)	Median OS in CRT and CT groups: 19.8 and 11.9 months, respectively (*p* < 0.0238)

#### 5.1.3. Definitive/Palliative Radiotherapy

For patients with unresectable IHCC or those who are not able to undergo major surgery, CT regimens such as gemcitabine/cisplatin with durvalumab or pembrolizumab are commonly used; however, survival remains limited (a median OS of approximately 12 months) and severe adverse effects from this regimen are common [30]. Thus, studies have investigated definitive and palliative RT for the treatment of unresectable IHCC.

Most notably, Hong et al. (Table 2) conducted a phase II clinical trial of 83 patients with primary unresectable (94%) or locally recurrent (6%) HCC (44), IHCC (37), or mixed IHCC-HCC (2) treated with passively scattered proton therapy to 67.5 GyE in 15 fx for peripheral tumors or 58.05 GyE in 15 fx for central tumors. Dose de-escalation was allowed to maintain a mean liver GTV of less than 24 GyE. A total of 61.5% of patients with IHCC additionally received CT. The median tumor size was 5.7 cm. The median OS in the IHCC group was 22.5 months and the two-year local control rate was 94.1%. Only patients receiving less than 60 GyE experienced local progression, suggesting a role for dose escalation [31]. Similarly, other studies have shown improved survival outcomes after proton therapy and with a greater RT dose (Table 2) [31,32,33].

Zhu et al. (Table 2) conducted a phase II clinical trial including 36 patients with primary unresectable (66.7%) or recurrent (33.3%) IHCC who received IMRT of at least 45 Gy (median 50 Gy) in fx of 2–2.5 Gy followed by anti-PD-1 immunotherapy (camrelizumab 200 mg every 3 weeks). The GTV included the primary tumor and positive nodes, and the CTV and PTV were additional 5 mm expansions. Most patients had stage III disease (91.7%), while a minority had stage II disease (8.3%). A total of 61.1% of patients had node-positive disease; 83.3% of patients had PD-L1-negative status, and 88.9% had a microsatellite-stable (MSS) status. At 19 months of follow-up, the median OS was 22 months, and the median progression-free survival (PFS) was 12 months. A complete response after treatment occurred in 11.1%, and 50% had a partial response [34]. Overall, these results show better survival than CT alone; however, the results must be interpreted with caution due to the small sample size [30,34].

In a retrospective study by Tao et al. (Table 2), 79 patients with inoperable IHCC who were treated with definitive IMRT (either photons or protons) to a median dose of 58.05 Gy in 3–30 fx for a median BED of 80.5 Gy were included. The GTV included the primary tumor and involved nodes, CTV was a 0–10 mm expansion, and the PTV was another 5 mm expansion from the CTV. For patients receiving > 50.4 Gy, a computerized axial tomography (CAT)- or fiducial-based kilovoltage image-guided inspiration breath hold was used; in other cases, an internal target volume (ITV) was created. Patients receiving more than 50.4 Gy also received a simultaneously integrated boost (SIB) to the GTV using a 0 to 5 mm PTV margin. A SIB of 75 Gy in 15 fx or 100 Gy in 25 fractures was used in select larger tumors for the GTV constricted by 1 cm. The tumor stage was 1, 2, 3, and 4 in 9%, 62%, 26%, and 3%, respectively. A total of 58% of patients had node-positive disease and 20% had metastatic disease. At a median follow-up of 33 months, the median OS was 30 months and 3-year OS was 44%. A higher BED was associated with improved local control (*p* = 0.009) and OS (*p* = 0.004), with patients receiving a BED of >80.5 Gy having a 3-year OS of 73% versus 38% for those receiving lower doses [35]. Like Zhu et al., this study shows a better survival compared to CT only, especially considering that a significant proportion of patients in this study had metastatic disease at the time of treatment [30,34,35]. Additionally, this study demonstrates a potential role for dose escalation given that a higher BED was associated with improved local control and OS.

Stereotactic body radiotherapy (SBRT) for unresectable IHCC has also been investigated [36,37]. Tse et al. (Table 2) conducted a phase I clinical trial investigating six-fx SBRT (a median dose of 36.0 Gy and a range of 24–54 Gy) for 41 patients with either unresectable HCC (31) or IHCC (10). Fluoroscopy and magnetic resonance imaging (MRI) were used to measure the liver breathing motion and liver position with exhaled breath holds. Abdominal compression was used for patients who could not perform a breath hold. GTV included the tumor and enhancing large-vessel thrombosis, an 8 mm margin was used to define the PTV, and an additional margin of at least 5 mm was used to define the CTV. All IHCC patients had advanced disease with either vascular involvement or extrahepatic disease at the time of treatment. Median OS for IHCC patients was 15 months. The one-year in-field local control rate was 65% for all patients. Two patients (20%) with IHCC developed transient biliary obstruction after the first few fx and two patients (20%) had a decline in liver function from Child–Pugh Class A to B within 3 months of treatment [36]. As the maximum tolerated dose was not determined from this study, better results may be achieved with a higher dose, warranting further study. Although the rate of developing biliary obstruction was high for IHCC patients, subsequent patients in this study were given pre-treatment dexamethasone and no new cases of obstruction were observed, suggesting that this may improve tolerability of the treatment. A meta-analysis also concluded that SBRT for the treatment of unresectable CCA leads to high rates of local control [38].

Válek et al. (Table 2) performed a prospective randomized trial in which 42 patients with malignant biliary strictures from advanced CCA received either a percutaneous stent followed by intraluminal Ir-192 brachytherapy (a median dose of 30 Gy) or a stent only. Median survival in the group receiving RT was 387.9 days and was 298 days in the group receiving a stent only. RT was also found to decrease the risk of stent obstruction (one patient in the RT group experienced obstruction, compared to four patients in the non-RT group) [39]. These data suggest that palliative RT for advanced CCA may improve the survival and efficacy of other palliative measures such as stenting, although the sample size is small. Indeed, two meta-analyses concluded that palliative RT for CCA leads to better survival with no difference in toxicities as compared to systemic therapy or stenting [40,41].

For patients with non-metastatic unresectable disease, the NCCN guidelines do not specify a preferred approach, but list systemic therapy, clinical trial participation, CRT with or without additional CT, arterially directed therapies, RT, referral for liver transplant, or supportive care as options. In the setting of metastatic disease, the NCCN guidelines recommend systemic therapy or clinical trial participation as the preferred approaches; however, arterially directed therapies and RT should also be considered [19].

**Table 2 curroncol-32-00545-t002:** Selected studies of non-operative management of IHCC that include RT.

Study	Type	*n*	Population	Treatment	Key Results
Bouras et al. (2002) [42]	Single-institution Retrospective	23	Patients with locally advanced CCA	RT (45–50 Gy, with a boost up to 60 Gy for R1 and R2 groups) +/− concurrent CT	Actuarial 1, 3, 5, year survival: 75%, 28%, 7%, respectively Median survival: 16.5 months
Válek et al. (2007) [39]	Prospective randomized trial	42 total; 21 received RT	Patients with malignant biliary strictures	Ir-192 brachytherapy (mean dose 30 Gy) + stent placement or stent placement only	Mean survival in RT group: 387.9 days Mean survival in no-RT group: 298 days
Tse et al. (2008) [36]	Prospective Phase I Clinical Trial	41 total; 10 with IHCC	Patients with unresectable HCC or IHCC	SBRT in 6 fx (median dose 36 Gy)	Median OS for IHCC patients: 15 months 20% of IHCC patients developed transient biliary obstruction
Kopek et al. (2010) [37]	Prospective	27	Patients with unresectable CCA	SBRT (45 Gy in 3 fx)	Median PFS: 6.7 months Median OS: 10.6 months
Hong et al. (2016) [31]	Prospective Phase II Clinical Trial	83 total; 37 with IHCC	Patients with unresectable HCC or IHCC	Proton therapy (58.0 GyE/15 fx for central tumors, 67.5 GyE/15 fx for peripheral tumors)	2-year local control for IHCC: 94.1% 2-year OS for IHCC: 46.5%
Tao et al. (2016) [35]	Single-institution Retrospective	79	Patients with unresectable IHCC	Photon or proton RT (median dose 58.05 Gy)	Median OS: 30 months 3-year OS: 44% Higher RT dose correlated with improved local control (*p* = 0.009) and OS (*p* = 0.004)
Smart et al. (2020) [33]	Single-institution Retrospective	66; 51 were treated with definitive intent	Patients with unresectable/locally recurrent IHCC	Hypofractionated RT (median dose 58.05 Gy), delivered in 15 fx	2-year OS and local control for patients treated with definitive intent: 62% and 93%, respectively Trend towards improved survival seen with proton therapy (HR 0.5, *p* = 0.05)
Parzen et al. (2020) [32]	Prospective	63 total; 25 with IHCC	Patients with unresectable HCC or IHCC	Hypofractionated proton RT (median dose 58.05 GyE)	1-year local control for IHCC: 90.9% 1-year OS for IHCC: 81.8% Patients receiving a BED > 75.2 Gy had better local control
Zhu et al. (2024) [34]	Prospective Phase II Clinical Trial	36	Patients with unresectable IHCC	RT (at least 45 Gy in 2–2.5 Gy/fx) followed by Anti-PD-1 therapy	1-year PFS: 44.4% Median PFS: 12 months Median OS: 22 months

### 5.2. Hilar/Perihilar Cholangiocarcinoma

pCCA accounts for about 50–70% of CCA cases [43]. In the United States, the incidence of eHCC (pCCA and dCCA) has remained approximately stable: 0.95 cases per 100,000 in 1973 versus 1.02 cases per 100,000 in 2012 [18]. pCCA often presents with biliary obstruction when the tumor is already at an advanced stage. High-quality imaging of the biliary tract system is necessary before any treatment as the extent of disease can be challenging to determine after treatment initiation [1,19]. As complete resection with negative margins is considered the only curative approach, surgical intervention with possible prior laparoscopic staging and/or biliary drainage should be performed in patients with resectable disease prior to other cancer therapies. For pCCA, surgery includes formal bile duct resection, major hepatic resection with likely caudate lobe resection, and an en bloc porta hepatis lymphadenectomy [1,15,19].

#### 5.2.1. Adjuvant Radiotherapy

As with other CCA sites, positive surgical margins and lymph node involvement have been associated with worse survival following surgery [1,15,16]. Thus, there has been investigation into adjuvant RT to augment local control.

SWOG S0809 (Table 3) was a phase II clinical trial investigating the delivery of adjuvant capecitabine and gemcitabine followed by RT with concurrent capecitabine in patients with pT2-4 or node-positive EHCC or GBC. RT was delivered as 54 Gy in 30 fx using 3D conformal RT (3D-CRT) (59.4 Gy in 33 fx could also be used for R1 resections) or was delivered as 52.5 Gy in 25 fx using IMRT (55 Gy in 25 fx could also be used for R1 resections). Most patients (86%) were treated with free breathing. A total of 79 patients were included, with 68% of them having EHCC. RT was delivered per protocol in 85% of patients. Two-year OS was 67% and the median OS was 35 months. Local recurrence rates were estimated to be 11% at 2 years and were found to be significantly higher in patients who had a deviation from the RT protocol (42% versus 11%, *p* = 0.02). Two-year OS and the local recurrence rate specific for EHCC was 68% and 13%, respectively [44]. Overall, this suggests that adjuvant CT followed by CRT may result in better survival and recurrence outcomes compared to surgery alone (2-year historical OS without RT estimated to be 55% and 38% for patients with R0 and R1 resections, respectively) for patients with pCCA [44]. However, results must be interpreted with caution as this study is not specific to pCCA.

Leng et al. (Table 3) conducted a study of 1917 pCCA patients from the Surveillance, Epidemiology, and End Results (SEER) registry, of which 762 patients received adjuvant RT. Patients in the group that received RT were more likely to have advanced disease (T3 or T4 tumor stage or node-positive) compared to the group who did not receive RT (86% versus 76.7%). The median OS in the adjuvant RT group was 23 months compared to 22 months in the non-RT group. After successful propensity score matching, the outcomes remained similar; the median OS for the adjuvant RT group was 22 months versus 23 months in the surgery-only group [45]. Panettieri et al. (Table 3) conducted a study utilizing the NCDB to analyze patients with resectable pCCA and showed that even for patients with R1 resection, adjuvant CRT did not improve survival compared to no adjuvant treatment (median OS 21.4 versus 19.8 months, *p* = 0.925) [46]. These studies both suggest no benefit of adjuvant RT, but must be interpreted with caution as the data are of a low quality due to the known confounders of these databases.

A retrospective study by Nassour et al. (Table 3) included 1846 pCCA patients from the NCDB, of which 57% received no adjuvant therapy, 31% received adjuvant CRT, and the remaining 12% received adjuvant CT only. A greater proportion of patients receiving adjuvant therapy had advanced T and N stages and a positive margin status, but propensity score matching was used in the data analysis. The median OS in the adjuvant therapy group was significantly improved compared to the no adjuvant therapy group (29.5 versus 21 months, HR 0.73, 95% CI 0.64–0.83), and the survival benefit was particularly pronounced in patients with positive resection margins (HR 0.53, 95% CI 0.42–0.67). Adjuvant CRT was associated with a better OS than adjuvant CT (31 months versus 25 months, *p* = 0.04) [47]. Similarly, Im et al. (Table 3) performed a retrospective study of 196 patients that found that adjuvant CRT or CT was associated with improved survival following resection of pCCA compared to no adjuvant therapy, but only adjuvant CRT showed a benefit for patients with an R1 resection [48]. These data suggest a benefit of adjuvant RT, particularly for patients with positive margins, but the data are of a low quality.

Studies of adjuvant RT for pCCA have, overall, showed mixed results (Table 3), but data specific to pCCA are, overall, of a low quality and SWOG S0809 includes dCCA and GBC patients as well; thus further study would be informative. For patients with resected pCCA, the current NCCN guidelines recommend either systemic therapy (usually Gemcitabine-based regimens) or clinical trial participation as the preferred options for adjuvant treatment, but lists CRT as an option for patients with R0 disease and negative nodes and lists CRT and CT and CRT as options to consider for patients with positive margins or positive nodes [19].

#### 5.2.2. Neoadjuvant Radiotherapy

As many cases of pCCA are not resectable, but operative management is considered the only curative approach, neoadjuvant RT may be useful for tumor downstaging. A neoadjuvant approach, the Mayo Protocol, consisting of external beam RT (EBRT) (45 Gy in 1.5 Gy twice dailyfx) with concurrent 5-FU for the first three days, followed by Ir-192 brachytherapy (20–30 Gy), followed by either 5-FU or capecitabine, followed by liver transplantation, has been used for selected patients for many years. In a study by Heimbach et al. (Table 3), 56 patients with unresectable stage I or II pCCA received the Mayo Protocol. Of the enrolled patients, (7.1%) patients died and (7.1%) had disease progression prior to the completion of neoadjuvant therapy. Of the 48 remaining patients, 14 (25%) had disease findings not allowing a transplant. Of the 28 patients who underwent transplantation, 3 patients died from perioperative complications (10.7%) and 4 developed recurrences between 22 months and 63 months post-transplantation. Seven patients (25%) had no residual disease identified at transplantation. The actuarial 5-year survival for all patients in this study was 54%, but was 82% post-transplantation for patients who were able to undergo surgery [49]. Similar to studies of neoadjuvant therapy for IHCC, these data suggest that many patients may not be able to undergo surgery after neoadjuvant therapy, but those who are able to undergo transplantation likely gain a significant survival benefit. Multiple subsequent studies (Table 3) have shown that neoadjuvant protocols similar to the Mayo Protocol have allowed for excellent survival outcomes in those who are able to undergo transplantation [50,51,52,53].

For patients with initially unresectable pCCA, the NCCN guidelines recommend biliary drainage if indicated and the consideration of referral to a liver transplant center. After that, recommended options include systemic therapy, clinical trial participation, CRT, RT, and CT, or palliative RT, with no option specified as being preferred [19].

**Table 3 curroncol-32-00545-t003:** Selected studies of operative management of pCCA that include RT.

Study	Type	*n*	Population	Treatment	Key Results
Heimbach et al. (2004) [49]	Prospective	56	Patients with unresectable stage I or II pCCA	EBRT (45 Gy in 1.5 Gy twice daily fx) + 5-FU → brachytherapy (20 Gy) → Capecitabine → Liver transplantation	3/28 patients who underwent transplantation died from perioperative complications 5-year actuarial survival for all patients: 54% 5-year actuarial survival for post-transplant patients: 82%
Rea et al. (2005) [50]	Single-institution Retrospective	125 total; 71 received RT	Patients with hilar CCA	RT and CT followed by liver transplantation or resection only	5-year OS in transplantation group was 82% vs. 21% in resection-only group
Murad et al. (2012) [51]	Multi-institution Retrospective	287	Patients with pCCA	EBRT + Brachytherapy + CT → +/−Liver transplantation	71 patients dropped out before surgery Intent-to-treat 2-year OS: 68% Post-transplant 2-year local control: 78%
Mukewar et al. (2015) [53]	Single-institution Retrospective	40	Patients undergoing biliary brachytherapy prior to liver transplantation for hilar CCA	Biliary high-dose-rate (HDR) brachytherapy via endoscopically placed nasobiliary tube (NBT) (9.3–16 Gy) + external RT (45 Gy) + CT → Liver transplantation	NBT/brachytherapy displacement observed in 20% of patients NBT kinking in 2.5% of patients NBT is technically feasible and reasonably safe
Ben-Josef et al. (2015) [44]	Prospective Phase II Clinical Trial	79	Stage pT2-4 or N+ or positive margins, M0 EHCC or GBC	Surgery → Gemcitabine + Capecitabine → Capecitabine + RT	2-year OS: 67% Median OS: 35 months
Nantajit et al. (2016) [26]	Single-institution Retrospective	27 total; 14 received CRT	Patients with resectable CCA	Surgery → +/− CRT or CT	2-year OS: 14 months 2-year OS in patients receiving > 54 Gy: 57.1% 2-year OS in patients receiving < 54 Gy: 23.1%
Leng et al. (2017) [45]	Database Retrospective	1917 total; 762 received RT	Patients with resectable pCCA	Surgery → +/− RT	Median OS in RT and non-RT group: 23 and 22 months, respectively (*p* = 0.651) In a matched population, adjuvant RT still did not show improved OS or cancer-specific survival
Nassour et al. (2018) [47]	Database Retrospective	1846 total; 793 received adjuvant therapy (some of which received RT)	Patients with resectable pCCA	Surgery → CT or CRT	CRT resulted in a slightly improved survival compared to CT (median OS 31 vs. 25 months) 5-year OS 31% for RT group and 23% for CT group
Krasnick et al. (2018) [54]	Multi-institution Retrospective	249 total, of which 94 received CRT	Patients with resectable hilar CCA	Surgery → RT, CT, or CT+RT	Adjuvant therapy was associated with improved OS (HR 0.58, *p* = 0.013), but this effect disappeared when node-positive patients were excluded (HR 0.76, *p* = 0.26)
Mukai et al. (2019) [27]	Single-institution Retrospective	32	Patients with resectable CCA	Surgery → RT (median dose 50 Gy)	2-year OS: 72.4% 2-year disease-free survival: 47.7% Median OS: 40 months
Wong et al. (2019) [28]	Prospective	18	Patients with locally advanced hilar CCA or IHCC planning to undergo liver transplantation	SBRT (40 Gy/5 fx) → CT → Liver transplantation	11 patients did not complete neoadjuvant protocol due to progression or severe toxicity 1-year post-transplant survival: 80% Median OS in dropout patients: 14.4 months
Zaborowski et al. (2020) [52]	Prospective	37	Patients with unresectable hilar CCA planning to undergo liver transplantation	7.5 Gy single-dose brachytherapy + 45–55 Gy EBRT + 5-FU + Capecitabine → +/− Liver transplantation	11 did not undergo surgery due to progression R0 rate: 96% pCR rate: 62% Median OS: 53 months 5-year OS: 55% Median OS in patients with pCR: 83.8 months (vs. 20.9 months for other patients)
Im et al. (2021) [48]	Single-Institution Retrospective	196 total; 39 received RT	Patients with resected hilar CCA	Surgery → CT, RT, CRT, or no adjuvant therapy	5-year OS: 32% CT and CRT were associated with better OS
Rodriguez et al. (2022) [29]	Database Retrospective	875 total; 57 received RT	Patients with resected CCA	Surgery → CT or CRT	Median OS in CRT and CT groups: 19.8 and 11.9 months, respectively (*p* = 0.0238)
Shridhar et al. (2023) [55]	Database Retrospective	1478	Patients with resected EHCC	Surgery → CT or CT+RT	Median OS for RT and non-RT patients: 34 and 27 months, respectively 5-year OS: 33% and 24%, respectively
Hoogwater et al. (2023) [56]	Multi-institution Retrospective	49 total; 27 received RT	Patients with pCCA undergoing liver transplantation	+/− neoadjuvant CRT → Liver transplantation	1-year OS in CRT and non-CRT groups: 65% and 91%, respectively Patients receiving neoadjuvant CRT had less risk of recurrence (HR 0.3) but greater risk of hepatic vascular complications (*p* = 0.045)
Gholami et al. (2023) [57]	Prospective Phase II Clinical Trial	69	Stage pT2-4 or N+ or positive margins, M0 EHCC, or GBC who completed 4 cycles of CT and RT	Surgery → Gemcitabine + Capecitabine → Capecitabine + RT	2-year OS: 70.6% and 60.9% for N0 and N+ disease, respectively 2-year DFS: 62.5% and 49.8% for N0 and N+ disease, respectively
Panettieri et al. (2024) [46]	Database Retrospective	1756 total; 49 received RT	Patients with pCCA	+/− Surgery → +/− CRT	Of patients who underwent R1 resection, CRT did not improve OS (21.4 vs. 19.8 months, respectively, *p* = 0.925)
Dominguez et al. (2024) [58]	Database Retrospective	4997 total; 469 with adjuvant CRT	pT2-4, pN0-1, M0 GBC or EHCC	Surgery → CT, CRT or none	Median OS for CRT: 36.9 months Survival after CT compared to CRT: HR 0.86 (*p* = 0.004)

#### 5.2.3. Definitive/Palliative Radiotherapy

Given that pCCA is often diagnosed when disease is advanced, studies have investigated the efficacy of definitive and palliative RT regimens. Autorino et al. (Table 4) conducted a phase II trial of 27 patients with unresectable EHCC, including pCCA (66.6%). Patients received EBRT (50.4 Gy to tumor and 39.6 Gy to nodes) and concurrent gemcitabine, with or without a HDR Ir-192 intraluminal brachytherapy boost (delivered trans-hepatically or by endoscopic retrograde) prescribed at 1 cm from the center of the source to 15–20 Gy. A total of 19.2% of patients had T2 disease, 38.4% had T3 disease, and 42.4% had T4 disease. A total of 40.7% of patients were node-positive at diagnosis. The overall median survival was 14 months, but the patients receiving the brachytherapy boost had a median OS of 21 months compared to 14 months for patients who did not receive a boost, suggesting the benefit of dose escalation. Similarly, local control was better for patients receiving brachytherapy (2-year local control 53% vs. 25%) [59]. A study demonstrated a median OS of 16 months for 92 patients receiving palliative CT only for unresectable pCCA, suggesting that the regimen in Autorino et al. likely did not result in improved survival compared to standard systemic therapy [59,60].

On the other hand, Liu et al. (Table 4) performed a retrospective study of 37 patients with unresectable pCCA in which patients either received EBRT (50–60 Gy in 2 Gy fx) and biliary drainage (70.3%) or biliary drainage only (29.7%), which showed that patients receiving RT had improved survival. The GTV included the tumor and any involved nodes. The PTV was an additional 5–10 mm expansion. Methods for respiratory motion management were used and 19.2% of patients experienced a pathologic complete response (pCR), 46.2% experienced a partial response, 23.1% had stable disease, and 11.5% had disease progression. The median OS in the group receiving RT was 22.8 months compared to 11.3 months in the biliary drainage only group [61]. Kaiser et al. (Table 4) demonstrated similar results with patients with unresectable pCCA receiving intraoperative RT (IORT). Themedian OS was 23.3 months compared to 9.4 months for patients receiving palliative surgery only [62].

Ishii et al. (Table 4) conducted a prospective study of 25 patients with unresectable EHCC that showed that palliative EBRT (30–50 Gy) combined with intraluminal brachytherapy (24–40 Gy) resulted in the full patency of the biliary ducts in 76% of patients, allowing for patients to live without biliary drainage tubes [63].

Overall, existing studies on definitive and palliative RT for the treatment of pCCA are small, results are conflicting, and associated toxicities are difficult to distinguish from disease effects/progression (Table 4). Further study is needed. For patients with unresectable disease, the NCCN guidelines recommend biliary drainage if needed followed by systemic therapy, clinical trial participation, CRT, CT, and RT, or palliative RT. In the metastatic setting, systemic therapy or clinical trial participation are preferred after biliary drainage, if required [19].

**Table 4 curroncol-32-00545-t004:** Selected studies of non-operative management of pCCA that include RT.

Study	Type	*n*	Population	Treatment	Key Results
Bouras et al. (2002) [42]	Single-institution Retrospective	23	Patients with locally advanced CCA	RT (45–50 Gy, with a boost up to 60 Gy for R1 and R2 groups) +/− concurrent CT	Actuarial 1-, 3-, and 5-year survival: 75%, 28%, and 7%, respectively Median OS: 16.5 months
Lu et al. (2002) [64]	Prospective Phase I/II Clinical Trial	18	Patients with unresectable EHCC	EBRT (45 Gy) + Ir-192 brachytherapy (7 Gy at 1 cm depth) per treatment	Median OS: 12.2 months 2-year survival: 27.8% Dose response suggested by improved survival with increasing brachytherapy dose
Ishii et al. (2004) [63]	Prospective	25	Patients with unresectable hilar or distal CCA	EBRT (30 or 50 Gy) combined with intraluminal brachytherapy (24–40 Gy) → removal of biliary drainage tube	In 76% of patients, full patency was achieved Median tube-free survival time: 76 days; 8 patients died tube-free Median OS: 9.3 months
Golfieri et al. (2006) [65]	Prospective	26	Patients with unresectable hilar CCA	Ir-192 brachytherapy +/− EBRT, biliary drainage, plastic endoprosthesis, or metallic stent placement and CT	Mean survival among multimodality patients: 10 months Median survival for brachytherapy-only patients: 6 months
Válek et al. (2007) [39]	Prospective randomized trial	42 total; 21 received RT	Patients with malignant biliary strictures	Ir-192 brachytherapy (mean dose 30 Gy) + EBRT (mean dose 50 Gy) + stent placement or stent placement only	Mean survival in RT group: 387.9 days Mean survival in no-RT group: 298 days
Kaiser et al. (2008) [62]	Single-institution Retrospective	18 total; 9 received RT	Patients with unresectable hilar CCA	IORT or palliative surgery	Median survival time in IORT and surgery groups: 23.3 months and 9.4 months, respectively
Kopek et al. (2010) [37]	Prospective	27	Patients with unresectable CCA	SBRT (45 Gy in 3 fx)	Median PFS: 6.7 months Median OS: 10.6 months
Kim et al. (2015) [66]	Single-institution Retrospective	25	Patients with recurrent EHCC or ampullary cancer	RT (median 54 Gy) +/− concurrent CT	2-year local control: 44% 2-year OS: 55%
Autorino et al. (2016) [59]	Prospective Phase II Clinical Trial	27	Patients with unresectable EHCC	Gemcitabine with concurrent EBRT (50.4 Gy to tumor, 39.6 Gy to nodes), some patients received a boost with Ir-192 brachytherapy	2-year local control: 29% 2-year OS: 27% Median OS: 14 months Median OS for patients receiving brachytherapy: 21 months 2-year local control for brachytherapy patients: 53%
Liu et al. (2016) [61]	Single-institution Retrospective	37	Patients with unresectable hilar CCA	Biliary drainage + 3D-CRT or biliary drainage only	RT response rate: 65.4% Median OS in RT group: 22.8 months (vs 11.3 months in non-RT group, *p* = 0.001)

### 5.3. Distal Cholangiocarcinoma

dCCA represents about 30% of CCA cases and often presents with non-specific symptoms including jaundice, vague abdominal pain, and weight loss, but can also present with biliary obstructive symptoms [67]. As with other CCA sites, if surgical resection is possible, this is currently considered the only potentially curative treatment approach. The current surgical approach of choice for dCCA treatment is a pancreatoduodenectomy (Whipple procedure) [15,67]. Following surgery, the estimated 5-year survival is about 37% and positive margins, lymph node involvement, vascular or pancreatic invasion, and a larger tumor size have been associated with worse outcomes [68].

#### 5.3.1. Adjuvant Radiotherapy

Kamarajah et al. (Table 5) performed a matched retrospective analysis utilizing the NCDB of 1509 dCCA patients receiving adjuvant RT after a pancreatoduodenectomy and 1509 patients not receiving RT. Patients receiving any kind of neoadjuvant therapy were excluded. Most patients (67.8% in the no-RT group and 67.3% in the RT group) had a tumor stages 2-3. A total of 51.2% of patients in the no-RT group had node-positive disease compared to 51.8% of patients in the RT group. Most patients, 71.6%, in both groups received adjuvant CT. Patients receiving adjuvant RT were found to have a significantly longer survival (median OS of 29.3 versus 26.8 months, *p* < 0.001). This remained significant when performing a stratified analysis by the nodal and marginstatus [69]. Although this study concluded that adjuvant RT improves survival, the observed difference must be interpreted with caution due to known confounders of this database.

Hou et al. (Table 5) conducted a multi-institutional retrospective study of 245 patients who underwent curative-intent resection for dCCA. A total of 43.7% of patients received adjuvant CT and RT, 17.6% received adjuvant CT only, and 38.8% of patients received no adjuvant therapy. A greater proportion of patients receiving adjuvant therapy had tumor stage T3 or T4 disease, lymph node involvement, and positive margins compared to the patients who received no adjuvant therapy (68.9% versus 31.1%, 61.3% versus 30.5%, and 68.5% versus 31.5%, respectively). Median OS in the no adjuvant therapy group was 24.5 months compared to 25.5 months in the adjuvant therapy group (*p* = 0.27). Adjuvant therapy was associated with a survival benefit for patients with lymph node involvement (median OS 20 versus 17.8 months, *p* = 0.03) or PNI (median OS 25.9 months versus 17.8 months, *p* = 0.03) [70]. These data suggest that adjuvant therapy, including RT, may be useful for patients with high-risk disease features.

Overall, studies focusing solely on dCCA currently do not show strong evidence for adjuvant RT. However, as there are not much existing data and no prospective data, further study is needed (Table 5). For patients with negative margins and negative nodes, the NCCN guidelines recommend systemic therapy or clinical trial participation as the preferred modality of adjuvant therapy, and also list CRT or observation as options. For patients with positive margins or nodes, systemic therapy and clinical trial participation are still the preferred approaches for adjuvant treatment, and CRT or CT and CRT can also be considered [19].

#### 5.3.2. Neoadjuvant Radiotherapy

Similar to other CCA sites, many dCCA patients present with unresectable disease. As surgical intervention is considered the only potential curative approach, neoadjuvant RT may help allow advanced tumors to become resectable. To our knowledge, there has been limited investigation of neoadjuvant RT for dCCA. Czito et al. (Table 5) performed a phase I clinical trial of patients with unresectable pancreatic cancer or dCCA who received Eniluracil/5-FU concurrently with RT (45 Gy followed by 5.4 Gy reduced fields), followed by potential surgery 4 weeks later. Three of the patients with dCCA were able to undergo surgery, of which negative margins were able to be achieved in one case [71]. No information on recurrence and survival outcomes was provided for these patients. Similar to other CCA sites, these data suggest that after neoadjuvant therapy, few patients are likely able to undergo R0 surgery. Kobayashi et al. (Table 5) conducted a retrospective study comparing 27 patients who received neoadjuvant CRT to 79 patients who received no neoadjuvant therapy prior to surgery. Patients all had an unresectable BTC; 63% of patients in the CRT group and 70% of patients in the non-CRT group had dCCA. The three-year recurrence-free survival (RFS) in the CRT group was 78.3% versus 56.8% in the non-CRT group (HR: 0.324, 95% CI: 0.12–0.88). However, the difference between OS was not significant (3-year OS 85% in the CRT group and 69% in the non-CRT group, *p* = 0.1583) [72]. These data suggest a lower risk of disease recurrence following neoadjuvant CRT, but must be interpreted with caution due to the retrospective nature of this study, the small sample size receiving CRT, and the fact that not all included patients had dCCA.

Overall, there are very limited data on neoadjuvant RT for dCCA, warranting further study (Table 5). The NCCN guidelines currently recommend surgery as the first treatment if resection is possible, and if initial surgery is not possible, recommends systemic therapy, clinical trial participation, CT and CRT, CRT, or palliative RT with the subsequent reconsideration of surgery. A preferred approach is not specified [19].

**Table 5 curroncol-32-00545-t005:** Selected studies of operative management of dCCA that include RT.

Study	Type	*n*	Population	Treatment	Key Results
Czito et al. (2006) [71]	Prospective Phase I Clinical Trial	13 total	Patients with resectable or locally advanced pancreatic adenocarcinoma or dCCA	Eniluracil/5-FU concurrently with RT (45 Gy followed by 5.4 Gy reduced fields) → Surgery	1 out of 3 patients with CCA who underwent surgery achieved negative margins
Ben-Josef et al. (2015) [44]	Prospective Phase II Clinical Trial	79	Stage pT2-4 or N+ or positive margins, M0 EHCC or GBC	Surgery → Gemcitabine + Capecitabine → Capecitabine + RT	2-year OS: 67% Median OS: 35 months
Nantajit et al. (2016) [26]	Single-institution Retrospective	27 total; 14 received CRT	Patients with resectable CCA	Surgery → +/− CRT or CT	2-year OS: 14 months 2-year OS in patients receiving > 54 Gy: 57.1% 2-year OS in patients receiving < 54 Gy: 23.1%
Kobayashi et al. (2017) [72]	Single-institution Retrospective	106 total; 27 received CRT	Patients with BTC (63% of CRT group had dCCA)	+/− CRT → Surgery	3-year RFS: 78% and 58% in CRT and non-CRT groups, respectively (*p* = 0.026)
Mukai et al. (2019) [27]	Single-institution Retrospective	32	Patients with resectable CCA	Surgery → RT (median dose 50 Gy)	2-year OS: 72.4% 2-year DFS: 47.7% Median OS: 40 months
Kamarajah et al. (2021) [69]	Database Retrospective	3018 total; 1509 received adjuvant RT	Patients with resected dCCA	Surgery → +/− RT	Median OS in RT and non-RT group: 29.3 and 26.8 months, respectively (HR = 0.86, *p* < 0.001)
Rodriguez et al. (2022) [29]	Database Retrospective	875 total; 57 received RT	Patients with resected CCA	Surgery → CT or CRT	Median OS in CRT and CT groups: 19.8 and 11.9 months, respectively (*p* = 0.0238)
Shridhar et al. (2023) [55]	Database Retrospective	1478	Patients with resected EHCC	Surgery → CT or CT+RT	Median OS for RT and non-RT patients: 34 and 27 months, respectively 5-year OS: 33% and 24%, respectively
Hou et al. (2023) [70]	Database Retrospective	245 total; 107 received RT	Patients with resected dCCA	Surgery → CT, CRT, or no adjuvant therapy	Adjuvant therapy was found to only improve survival for node-positive patients and those with PNI Median OS for PNI patients: 25.9 months for adjuvant therapy vs. 17.8 months for surgery alone Median OS for node-positive patients: 20 months for adjuvant therapy vs. 17.8 months for surgery alone
Gholami et al. (2023) [57]	Prospective Phase II Clinical Trial	69	Stage pT2-4 or N+ or positive margins, M0 EHCC or GBC who completed 4 cycles of CT and RT	Surgery → Gemcitabine + Capecitabine → Capecitabine + RT	2-year OS: 70.6% and 60.9% for N0 and N+ disease, respectively 2-year DFS: 62.5% and 49.8% for N0 and N+ disease, respectively
Dominguez et al. (2024) [58]	Database Retrospective	4997 total; 469 with adjuvant CRT	pT2-4, pN0-1, M0 GBC or EHCC	Surgery → CT, CRT, or none	Median OS for CRT: 36.9 months Survival benefit of CRT compared to CT: HR 0.86

#### 5.3.3. Definitive/Palliative Radiotherapy

Similar to other CCA sites, many patients with dCCA may have unresectable disease or not be able to undergo major surgery required for curative intent. In these scenarios, definitive or palliative RT may be useful in improving local control and alleviating symptoms. Preferred systemic therapy regimens for patients with unresectable or metastatic disease include the combination of gemcitabine/cisplatin with immunotherapy such as Pembrolizumab [19]. To our knowledge, there are limited existing studies focusing solely on definitive/palliative RT for dCCA. Laurent et al. (Table 6) conducted a prospective phase I-II trial that included 17 patients with advanced pancreatic cancer and 5 patients with advanced dCCA that received two cycles of gemcitabine/oxaliplatin followed by RT to a dose of 45 Gy in 25 daily fx with concurrent gemcitabine/oxaliplatin. The median time to progression and OS was 8 months and 17 months, respectively [73]. Given that the median survival for unresectable pancreatic cancer and dCCA is less than 10 months, this study suggests that RT may improve the OS for dCCA patients [74,75,76]. However, this is a small study and most patients had pancreatic cancer rather than dCCA.

A retrospective study by Kim et al. (2015) included 25 patients who underwent salvage RT for recurrent BTC, of which 15 had dCCA, 7 had pCCA, and 3 had ampullary cancer. 3D-CRT was used to a dose of 44–45 Gy to the CTV (a 0.5–1 cm expansion from the GTV) at a 1.8–2 Gy per fraction. An additional 9–10 Gy in five fx was boosted to the GTV. A total of 76% of patients received concurrent CT. At a median follow-up of 16 months, 36% of patients experienced local recurrence and 44% experienced distant recurrence. The two-year OS was 55% with a median OS of 24 months. The median OS was significantly longer in patients receiving CRT compared to RT alone (27 versus 8 months, *p* = 0.043) [66]. These data are encouraging, and especially for CRT in the definitive/salvage therapy setting.

Selected studies on the non-operative management of dCCA including definitive or palliative RT are summarized in Table 6. As many studies combine EHCC sites, as discussed for pCCA, there are limited existing data on definitive and palliative RT and associated toxicities may be high. The NCCN guidelines list palliative RT, CRT, or CT and CRT as potential options for unresectable dCCA, along with systemic therapy and clinical trial participation [19].

**Table 6 curroncol-32-00545-t006:** Selected studies of non-operative management of dCCA that include RT.

Study	Type	*n*	Population	Treatment	Key Results
Bouras et al. (2002) [42]	Single-institution Retrospective	23	Patients with locally advanced CCA	RT (45–50 Gy, with a boost up to 60 Gy for R1 and R2 groups) +/− concurrent CT	Actuarial 1-, 3-, and 5-year survival: 75%, 28%, and 7%, respectively Median survival: 16.5 months
Lu et al. (2002) [64]	Prospective Phase I/II Clinical Trial	18	Patients with unresectable EHCC	EBRT (45 Gy) + Ir-192 brachytherapy (7 Gy at 1 cm depth) per treatment	Median OS: 12.2 months 2-year OS: 27.8% Dose response suggested by improved survival with increasing brachytherapy dose
Ishii et al. (2004) [63]	Prospective	25	Patients with unresectable hilar or distal CCA	EBRT (30 or 50 Gy) combined with intraluminal brachytherapy (24–40 Gy) → removal of biliary drainage tube	In 76% of patients, full patency was achieved Median tube-free survival time: 76 days; 8 patients died tube-free Median OS: 9.3 months
Válek et al. (2007) [39]	Prospective randomized trial	42 total; 21 received RT	Patients with malignant biliary strictures	Ir-192 brachytherapy (mean dose 30 Gy) + EBRT (mean dose 50 Gy) + stent placement or stent placement only	Mean survival in RT group: 387.9 days Mean survival in no-RT group: 298 days
Laurent et al. (2009) [73]	Prospective Phase I–II Clinical Trial	22 total; 5 with distal EHCC	Patients with unresectable pancreatic cancer or dCCA	Two cycles gemcitabine/oxaliplatin → RT (45 Gy in 25 fx)	Median OS: 17 months Median PFS: 8 months
Kopek et al. (2010) [37]	Prospective	27	Patients with unresectable CCA	SBRT (45 Gy in 3 fx)	Median PFS: 6.7 months Median OS: 10.6 months
Kim et al. (2015) [66]	Single-institution Retrospective	25	Patients with recurrent EHCC or ampullary cancer	RT (median 54 Gy) +/− concurrent CT	2-year local control: 44% 2-year OS: 55%
Autorino et al. (2016) [59]	Prospective Phase II Clinical Trial	27	Patients with unresectable EHCC	Gemcitabine with concurrent EBRT (50.4 Gy to tumor, 39.6 Gy to nodes), some patients received a boost with Ir-192 brachytherapy	2-year local control: 29% 2-year OS: 27% Median OS: 14 months Median OS for patients receiving brachytherapy: 21 months 2-year local control for brachytherapy patients: 53%

## 6. Radiotherapy for Gallbladder Cancer

GBC is rare overall, but represents about 50% of BTCs. In 2020, there were an estimated 115,949 new cases of GBC and 84,695 deaths worldwide due to the disease [17]. GBCs are known to be aggressive with a poor prognosis; 5-year survival is estimated to be less than 20%. Many patients have locally advanced or metastatic disease at the time of presentation [1,77]. The initial management of GBC depends on the resectability, metastatic status, and the way in which the patient was diagnosed. If an incidental finding suspicious for GBC is observed during surgery, intraoperative staging with or without a biopsy should be performed if possible. If the mass is resectable, surgical excision is recommended and involves cholecystectomy with en bloc hepatic resection, a lymphadenectomy, and a possible bile duct excision if there is suspected malignant involvement. If a mass suspicious for GBC is observed on imaging, further imaging workup, assessment of the hepatic reserve, and consideration of a staging laparoscopy should be considered. If resectable, surgery should then be performed. If GBC is found incidentally on review of pathology, further management depends on the staging. If it is T1a with negative margins, observation is the preferred next step. If it is T1a with positive margins or T1b, a further imaging workup and a staging laparoscopy should be considered. If resectable, surgery should be the next step. If a cystic duct node is positive in the pathology, a further imaging workup should be performed followed by neoadjuvant systemic therapy or clinical trial participation, followed by surgical intervention if possible. If GBC presents with jaundice, an imaging workup, biliary drainage, and staging laparoscopy should be considered. If the disease is resectable, neoadjuvant systemic therapy or clinical trial participation should be considered, followed by surgical intervention [19].

### 6.1. Adjuvant Radiotherapy

The risk of recurrence after surgical intervention has been found to be especially high for patients with lymph node involvement or positive surgical margins, suggesting a potential role for adjuvant RT [78,79]. Additionally, up to 75% of post-operative recurrences are in the potential target volume of adjuvant RT [80]. Patients with T2 or greater disease may also be at particular risk for locoregional recurrence, thus would benefit from adjuvant RT [81]. Given this, many studies have investigated the efficacy of adjuvant RT for GBC.

Ostwal et al. (Table 7) (GECCOR-GB Trial) conducted a parallel running phase II trial with one group receiving adjuvant gemcitabine/capecitabine and the other group receiving capecitabine with concurrent RT for 90 patients (45 in each arm) with stage II-III GBC with R0 or R1 resection. RT was delivered as 45 Gy in 25 daily fx. One-year and two-year DFS was 88.9% (95% CI 79.5–98.3%) and 74.8% (95% CI 60.4–89.2%) in the gemcitabine/capecitabine group, and 77.8% (95% CI 65.4–90.2%) and 74.8% (95% CI 59.9–86.3%) in the CRT group. There were no locoregional recurrences in the CRT arm, compared to 7% of patients in the gemcitabine/capecitabine arm. Two-year OS was 88% (95% CI 77.8–98.2%) in the gemcitabine/capecitabine arm and 80.6% (95% CI 68.2–93.0%) in the CRT arm. A third of patients in the gemcitabine/capecitabine group experienced a distant relapse, compared to 27% in the CRT group. The median OS was not reached for either arm, but when only patients with stage III disease were considered, the median OS was 26.7 months (95% CI, 25.7–27.6) in the gemcitabine/capecitabine group compared to 11.4 months (95% CI 3–19.8) in the CRT group [82]. Given that distant metastasis was the most common pattern of recurrence, it is possible that the CRT group would have benefited from additional CT leading to better outcomes. As the 1-year DFS in both arms was above the pre-specified trial end point, a larger comparative phase III trial is being planned and will be of interest [82].

As previously described, SWOG S0809 (Table 7) was a phase II clinical trial investigating the delivery of adjuvant capecitabine and gemcitabine followed by RT with concurrent capecitabine in patients with pT2-4 or node-positive EHCC or GBC. The two-year OS and local recurrence rate specific for GBC were 56% and 8%, respectively [44].

Selected studies investigating adjuvant RT are included in Table 7. Other than the data from SWOG S0809 and GECCOR-GB, the studies are retrospective. Most studies have suggested that adjuvant RT leads to improved disease control and survival; however, some exceptions include studies performed by Mantripragada et al. (Table 7) and Cai et al. (Table 7), which found no benefit to adjuvant RT [83,84]. Additionally, some studies have found adjuvant RT to only have a benefit for patients with advanced disease or high-risk disease features, such as lymph node involvement or positive margins [85,86,87,88,89]. As a study by Kresl et al. (Table 7) found that the 5-year OS for patients receiving adjuvant RT greater than 54 Gy was 100% (compared to 65% for patients receiving less than 54 Gy), the GBC response to RT may be dose-dependent; thus patients may benefit from dose escalation [90].

Several meta-analyses have been conducted on the efficacy of adjuvant RT for GBC. Choi et al. (2022) included 21 studies with a total of 6876 GBC patients. In studies where the RT and non-RT groups were clinically similar, the OS was better in the RT group (5-year OS 44.9% vs. 20.9%, OR = 1.92, *p* = 0.008). The locoregional recurrence rate was also better in the RT group (OR 0.21, *p* = 0.001), but no difference in the rate of distant metastasis was observed (OR 1.32, *p* = 0.332) [91]. Kim et al. (2018) included 14 retrospective studies with a total of 9364 patients and concluded that RT significantly reduced the risk of death and disease recurrence despite generally being given to patients with higher-risk disease characteristics. However, further analysis revealed that RT was only associated with survival benefit for patients with node-positive or R1 disease [92]. Ren et al. (2020) included 21 clinical trials and concluded that RT improves the OS and reduces LR for patients with EHCC and GBC, especially patients with node-positive disease or positive surgical margins [93].

Overall, there are limited prospective data investigating adjuvant RT for GBC; thus, further study is needed. The evidence is currently strongest for patients with high-risk disease features, such as higher disease stage or positive surgical margins. Indeed, a recent ASCO clinical guideline recommended CRT for patients with EHCC or GBC who have R1 resections [94]. The NCCN guidelines recommend adjuvant systemic therapy or clinical trial participation as the preferred approaches after surgery for patients with negative margins and regional nodes. Observation and CRT are also listed as potential options. For patients with positive margins or regional nodes, systemic therapy or clinical trial participation are still preferred, with a combination of CT and CRT or CRT alone also listed as possible choices [19].

### 6.2. Neoadjuvant Radiotherapy

Surgical intervention is currently considered to be the only curative approach for GBC patients, but many cases are inoperable, especially if diagnosed non-incidentally [1]. Thus, neoadjuvant RT may be useful in allowing patients with initially unresectable disease to be able to undergo curative surgery, with most existing studies showing a potential benefit (Table 7). Engineer et al. (Table 7) conducted a prospective study of 28 patients with stage III GBC who underwent concomitant gemcitabine and RT (helical tomotherapy to a dose of 57 Gy in 25 fx to the gross tumor and 45 Gy in 25 fx for the surrounding nodes) followed by surgery if R0 resection was felt to be possible. Most patients, 89.3%, were able to complete the neoadjuvant treatment protocol. A total of 71% of patients were felt to have a partial or complete radiological response. A total of 64% of patients underwent surgical exploration and R0 resection was able to be achieved in 56% of patients enrolled in the study. The median survival for all patients in the study was 24 months, compared to 47 months for patients able to undergo R0 resection [95]. Overall, these data suggest that neoadjuvant CRT allows for a significant proportion of patients with initially unresectable disease to undergo surgery, with a large survival benefit. However, one limitation of this study is the small sample size. Other prospective studies have found R0 resection rates of 15% (RT dose of 45 Gy) and 46% (RT dose of 55–57 Gy), suggesting that higher RT doses may be more effective, and thus there should be more investigation into dose escalation [96,97]. Of note, de Aretxabala et al. (Table 7) conducted a prospective study in which 23 GBC patients received neoadjuvant 5-FU with concurrent RT (45 Gy in 25 fx), and found that the actuarial survival of patients treated with CRT was worse than the controls who did not receive CRT [98]. However, the study sample size is small and the RT dose was only 45 Gy.

Hakeem et al. (2019) performed a systematic analysis of the role of neoadjuvant CT or CRT for GBC. A total of 474 patients from eight different studies were included, of which 16% received neoadjuvant CRT, with the remaining patients receiving CT. The study concluded that there were insufficient data to support neoadjuvant therapy for GBC as only 35.4% of patients were able to undergo R0 resection and 30.6% had progressive disease despite neoadjuvant therapy. However, the overall survival ranged from 18.5–50.1 months for those able to undergo curative resection compared to 5–10.8 months for those who were unable to, suggesting that there may be a large survival benefit. The resection and survival outcomes may have been better if a greater proportion of included patients received CRT rather than just CT [99]. Overall, further study of neoadjuvant therapy is needed given that no randomized data exist and existing prospective studies have small sample sizes (Table 7). NCCN guidelines currently recommend systemic therapy or clinical trial participation as the preferred options for patients with unresectable disease, with the subsequent reconsideration of resection. Palliative RT is also listed as an option. The only scenario in which neoadjuvant therapy is suggested is in patients who have GBC as an incidental finding on pathologic review and have a positive cystic node; however, systemic therapy rather than RT is currently recommended [19].

**Table 7 curroncol-32-00545-t007:** Selected studies of operative management of GBC that include RT.

Study	Type	*n*	Population	Treatment	Key Results
Kresl et al. (2002) [90]	Single-institution Retrospective	21	GBC patients with resectable disease	Surgery → CRT	5-year OS: 33% For patients with no residual disease, 5-year OS was 64% (vs. 0% for those with residual disease) 5-year OS was 100% for patients receiving > 54 Gy (vs. 65% for patients receiving < 54 Gy)
de Aretxabala et al. (2004) [98]	Prospective	23	Patients undergoing neoadjuvant CRT for GBC	CRT (5-FU + 45 Gy/25 fx) → +/− Surgery	Patients treated with CRT had worse actuarial survival than patients who did not receive CRT
Mojica et al. (2007) [85]	Database Retrospective	3187 total; 17% received RT	GBC patients with resectable disease	Surgery → +/− RT	RT group median survival: 14 months No RT median survival: 8 months Survival benefit only seen for patients with regional spread or infiltration of the liver
Hyder et al. (2014) [100]	Database Retrospective	5011 total; 899 received RT	GBC patients with resectable disease	Surgery → +/− RT	Median OS: 15 months RT was associated with better 1-year OS—HR 0.45—but not with 5-year OS—HR 1.06
Ben-Josef et al. (2015) [44]	Prospective Phase II Clinical Trial	79	Stage pT2–4 or N+ or positive margins, M0 EHCC or GBC	Surgery → Gemcitabine + Capecitabine → Capecitabine + RT	2-year OS: 67% Median OS: 35 months
Wang et al. (2015) [101]	Multi-institution Retrospective	112 total; 68 receiving RT	Patients undergoing extended surgery for GBC	Surgery → RT, CT, or no adjuvant therapy	5-year OS with adjuvant RT was 49.7% versus 52.5% for surgery only (*p* = 0.20) despite RT patients being more likely to have advanced disease Adjuvant RT associated with decreased local failure: HR 0.17
Hoehn et al. (2015) [102]	Database Retrospective	6690 total; 15.1% received CRT	GBC patients with resectable disease	Surgery → CT, CRT, or no adjuvant therapy	CRT was associated with improved survival (HR 0.77, 95% CI 0.66–0.90) especially for node-positive cases (HR 0.64, 95% CI 0.53–0.78)
Kim et al. (2016) [86]	Multi-institution Retrospective	291 total; 44 (15.1%) received adjuvant CRT	GBC patients with resectable disease	Surgery → CT, CRT, or no adjuvant therapy	CRT was associated with improved DFS (HR: 0.43) and OS (HR: 0.26) Only patients with high-risk disease factors benefitted from CRT (T3/4, or LN involvement or R1 status)
Mantripragada et al. (2016) [83]	Database Retrospective	4775 total; 13.5% received adjuvant CRT	GBC patients with resectable disease	Surgery → CT, CRT, or no adjuvant therapy (median RT dose 50.4 Gy)	Three-year OS: 39.9% Adjuvant therapy did not improve OS: HR 1.01
Engineer et al. (2016) [95]	Prospective	28	Patients with Stage III GBC undergoing neoadjuvant CRT	CRT (RT delivered as 57 Gy in 25 fx to gross tumor and 45 Gy in 25 fx to LNs) → Surgery	71% achieved partial or complete response to CRT radiologically 56% of patients in the study were able to undergo R0 resection Median OS was 20 months for all patients vs. 47 months for patients who underwent R0 resection
Agrawal et al. (2016) [96]	Prospective	40 total; 25 received CRT	Patients undergoing neoadjuvant CRT for locally advanced GBC	CRT (RT dose 45 Gy) or CT → +/− Surgery	Resectability rate: 15% (all achieved R0 resection) 16.6% and 83.3% rate of histopathological CR of liver and lymph nodes, respectively
Kasumova et al. (2017) [103]	Database Retrospective	6825 total; 1919 received adjuvant therapy	pT2/T3 GBC patients who have undergone resection	Surgery +/− adjuvant CT and/or RT	Median OS for patients undergoing extended cholecystectomy followed by CT+RT: 27.7 months (95% CI 21.9–33.6) compared to 15.9 months (95% CI 13.4–23.6) for CT after surgery alone
Gu et al. (2018) [104]	Database Retrospective	78 total; 39 received adjuvant CRT	pT2-T4, N0-1, MO GBC patients	Surgery → CRT (median RT dose 50 Gy) or no adjuvant therapy (matched)	Median survival for CRT group vs. non-CRT group: 27 vs. 13 months (*p* = 0.04); median DFS 23 months vs. 7 months (*p* = 0.004)
Kim et al. (2019) [87]	Database Retrospective	151 total; 98 received CRT	GBC patients with resectable disease	Surgery → CT, CRT, or no adjuvant therapy	In patients with T2–3 node-positive disease, CRT resulted in lesser local recurrence (11.4% vs. 78.1%) and distant recurrence (42.8% vs. 73.9%) at a median follow-up of 42.7 months
Kapoor et al. (2020) [105]	Single-institution Retrospective	36	GBC patients with poor performance status undergoing adjuvant CRT	Surgery → CRT	Median OS and RFS: 26 and 21 months, respectively
Choudhary et al. (2021) [106]	Single-institution Retrospective	50 total; 30 receiving RT or CRT	Stage II and III GBC who have undergone resection	Radical surgery followed by RT, CT, or CRT (RT to a dose of 40–54 Gy)	CRT mean OS, DFS: 44 months, 43.6 months RT mean OS, DFS: 12.5 months, 9.6 months CT mean OS, DFS: 15.1 months, 12.4 months
Lee et al. (2021) [88]	Multi-institution Retrospective	733 total; 146 received CRT	GBC patients who underwent curative intent surgery	Surgery → CT, CRT, or no adjuvant therapy	For stage III-IV cancers, but not stage II, CRT was associated with better 5-year LRFS (67.8% vs. 45.2% vs. 56.9%), RFS (56.9% vs. 37.9% vs. 28.8%), and OS (45.0% vs. 30.0% vs. 45.7%) than no-adjuvant or CT groups, respectively
Cai et al. (2021) [84]	Database Retrospective	726 total	Stage IIIb GBC patients who underwent resection	Surgery → +/− RT	RT failed to improve GBC-specific survival (median OS 22 months vs. 19 months in no adjuvant RT group)
Wan et al. (2021) [107]	Database Retrospective	2689 total; 542 received adjuvant CRT	Stage II-IV GBC patients who underwent resection	Surgery → CT, CRT, or no adjuvant therapy	CT or CRT did not improve OS in Stage II patients CRT was superior to CT, which was superior to no adjuvant therapy for Stage III and IV patients (*p* < 0.001)
Song et al. (2022) [108]	Database Retrospective	7866 total; 1225 receiving RT	Patients with GBC	No RT or surgery vs. surgery vs. palliative RT vs. surgery + adjuvant RT	Surgery + adjuvant RT median OS: 22 months Surgery only median OS: 16 months RT only median OS: 8 months No treatment median OS: 4 months
Kamarajah et al. (2022) [109]	Database Retrospective	4134 patients; 2067 received RT	GBC patients with resectable disease	Surgery → +/− RT	RT resulted in improved survival (OS 26.2 vs. 21.5 months, HR 0.82, *p* < 0.001)
Loyal et al. (2022) [97]	Phase II Clinical Trial	26	Patients undergoing neoadjuvant CRT for GBC	CRT → Surgery	77% of patients had a partial or complete response to neoadjuvant CRT; 46% had an R0 resection At median follow-up of 38 months, OS was 38%
Alam et al. (2023) [89]	Single-institution Retrospective	176 total; 49 received CT followed by CRT	GBC patients who underwent simple cholecystectomy and were not eligible for further surgery	Simple cholecystectomy → CT or CT followed by CRT	CT followed by CRT resulted in better survival for patients with residual disease in the gallbladder bed (*p* = 0.003), Median OS: 27 months in CRT group vs. 19 months in CT group for patients with residual disease after surgery
Gholami et al. (2023) [57]	Secondary Analysis of Prospective Phase II Clinical Trial	69	Stage pT2-4 or N+ or positive margins, M0 EHCC or GBC who completed 4 cycles of CT and RT	Surgery → Gemcitabine + Capecitabine → Capecitabine + RT	2-year OS: 70.6% and 60.9% for N0 and N+ disease, respectively 2-year DFS: 62.5% and 49.8% for N0 and N+ disease, respectively
Dominguez et al. (2024) [58]	Database Retrospective	4997 total; 469 with adjuvant CRT	pT2-4, pN0-1, M0 GBC or EHCC	Surgery → CT, CRT, or none	Median OS for CRT: 36.9 months Survival benefit of CRT compared to CT: HR 0.86^82^
Ostwal et al. (2024) [82]	Prospective Phase II Clinical Trial	90 total; 45 receiving adjuvant RT	Stage II or III GBC with R0 or R1 resection	Surgery → adjuvant gemcitabine + capecitabine or capecitabine + capecitabine/RT (CCRT)	1- and 2-year DFS for GC patients: 88.9%, 77.8% 1- and 2-year DFS for CCRT patients: 74.8%, 74.8%

### 6.3. Definitive/Palliative Radiotherapy

Few studies have investigated the use of definitive RT for GBC, possibly because of the poor prognosis of GBC without surgery. However, as many cases of GBC are not operable, and some patients may not be able to undergo major surgery due to medical comorbidities, definitive RT may represent the only possibility of curative-intent local intervention. Palliative RT may also be useful in reducing symptoms such as pain and jaundice, thus improving quality of life.

RACE-GB (Table 8) was a randomized clinical trial of 67 patients receiving only CT (gemcitabine/cisplatin) and 68 patients receiving consolidation CRT after initial CT for unresectable GBC. CRT was delivered using 3D-CRT to a dose of 45 Gy in 25 fx to the tumor and lymphatics followed by a boost of 9 Gy in 5 fx to the tumor. This was given with concurrent capecitabine. The consolidation CRT group demonstrated a better OS from the time of randomization, 10 months, compared to 4 months for the CT-only group (HR: 0.43; 95% CI 0.32–0.62). The median PFS from the time of randomization was 7 months in the CRT group and 1 month in the no-CRT group. The patient’s quality of life was measured over time, and no difference in the Functional Assessment of Cancer Therapy—Hepatobiliary Trial Outcome Index was found (*p* = 0.099) between the groups [110]. These data suggest that consolidation CRT improves disease control outcomes for patients with advanced GBC without worsening their quality of life.

Sinha et al. (Table 8) performed a retrospective study of 45 patients with non-metastatic, locally advanced, unresectable GBC, 25 of which were treated with definitive-intent RT (delivered to a total dose of 52–57 Gy) with concurrent gemcitabine (others received CT only: gemcitabine and either cisplatin or oxaliplatin). Two-year PFS was found to be 18.6% in the RT group, compared to 0% in the no-RT group (*p* = 0.0001). The two-year OS was 37.3% versus 4.3% in the same groups, respectively (*p* = 0.0001), and the median OS was 18 months versus 7.5 months. The locoregional failure rates were also significantly lower in the RT group at a median follow-up of 11.5 months (32% vs. 85%, *p* = 0.0002) [111]. Overall, these data suggest a significant survival benefit from definitive RT, but the study is retrospective and includes a limited number of patients. Similarly, studies by Verma et al. (Table 8) and Alam et al. (Table 8) also suggest a significant survival benefit from definitive RT [112,113]. A study by Singh et al. did not demonstrate a PFS benefit; however, the total RT dose delivered was lower compared to other studies (30–45 Gy) [114]. Further study is needed to determine the optimal dosing. Dose escalation via high-dose ablative therapy may be more effective as definitive therapy, and may be better achieved while limiting toxicities to organs at risk using proton therapy, warranting further investigation into this modality [79].

Song et al. (Table 8) conducted a study comparing outcomes of 7866 patients from the SEER database with GBC who received no RT or surgery (27.1%), surgery alone (57.3%), surgery followed by adjuvant RT (12.9%), or palliative RT only (2.7%). Palliative RT led to a median survival of 8 months compared to 4 months for no treatment (HR: 0.68, 95% CI 0.58–0.79) [108]. This suggests the potential benefit of palliative RT; however, this must be interpreted with caution due to the low data quality, relatively few patients receiving palliative RT, and no information provided about the dosing/fractionation of RT.

Overall, most studies have shown a potential benefit to palliative and definitive RT. NCCN guidelines currently recommend systemic therapy (typically gemcitabine/cisplatin with durvalumab or pembrolizumab) or clinical trial participation as the preferred options for patients with unresectable GBC, but palliative RT and best supportive care are also listed as options [19].

**Table 8 curroncol-32-00545-t008:** Selected studies of non-operative management of GBC that include RT.

Study	Type	*n*	Population	Treatment	Key Results
Eleftheriadis et al. (2001) [115]	Case report	1	Grade IV GBC	EBRT using 60co (30 Gy/10 fx)	No local tumor extension at 12 months of follow-up; performance status remained good
Singh et al. (2014) [114]	Single-institution Retrospective	50 total; 18 received adjuvant RT	Patients with unresectable GBC	Best supportive care or CT or CT+RT (delivered to 30–45 Gy)	Progression-free survival at 15 months was 38% vs. 30% vs. 18% for patients receiving CT+RT, CT, and best supportive care, respectively (*p* = 0.538)
Verma et al. (2018) [112]	Database Retrospective	1199 total; 327 received CRT	Patients with unresectable, nonmetastatic GBC	CT or CRT	Median OS in CRT vs. CT groups: 12.9 vs. 7.8 months (*p* = 0.001)
Song et al. (2022) [108]	Database Retrospective	7866 total; 209 receiving palliative RT	Patients with GBC	No RT or surgery vs. surgery vs. palliative RT vs. surgery + adjuvant RT	Surgery + adjuvant RT median OS: 22 months Surgery-only median OS: 16 months Palliative RT-only median OS: 8 months No treatment median OS: 4 months
Alam et al. (2022) [113]	Single-institution Retrospective	145; 35% of which received CRT	Patients with locally advanced GBC	CT → +/− CRT (RT to 45–54 Gy)	Median survival for CRT group: 14 months (vs. 7 months in non-CRT group, *p* = 0.04)
Sinha et al. (2022) [111]	Single-institution Retrospective	45 total; 25 received RT	Patients with unresectable GBC	CT or CRT	2-year OS CRT: 37.3% vs. 2-year OS CT: 5% (*p* = −0.0001) Rate of local progression (at 11.5 months): 85% in CT group, 32% in CRT group (*p* = 0.0002)
Agrawal et al. (2025) [110]	Prospective Randomized Clinical Trial	135; 68 received RT	Patients with unresectable GBC	CT → CRT or observation	Median OS for CRT vs. no CRT group: 10 months vs. 4 months (HR: 0.43, 95% CI: 0.32–0.62)

## 7. Radiotherapy for Ampullary Cancer

Ampullary cancers are rare malignancies occurring at the ampulla of Vater. Periampullary refers to cancers originating near the ampulla of Vater, including the head of the pancreas, distal bile duct, and duodenum. Only 5625 cases of ampullary cancer were diagnosed per the SEER database between 1973 and 2005 [116]. The 5-year survival for ampullary cancers is reported to be between 35 and 50% [117]. For patients with resectable disease at diagnosis, a pancreaticoduodenectomy is the preferred next step, but neoadjuvant systemic therapy with or without CRT should also be considered per the NCCN guidelines [118]. Endoscopic removal or an ampullectomy may also be considered [117].

### 7.1. Adjuvant Radiotherapy

Given the high rates of recurrence despite surgery, studies have investigated adjuvant RT for ampullary cancers. Klinkenbijl et al. (Table 9) conducted a phase III clinical trial published in 1999 investigating the efficacy of adjuvant RT with concurrent 5-FU for patients with T1-2N0-1aM0 cancer of the pancreatic head (55%) or T1-3N0-1aM0 periampullary cancer (45%). A total of 103 patients were assigned to the treatment group and 104 patients were assigned to no adjuvant treatment. RT was delivered as 20 Gy in 2 Gy fx for two weeks, followed by a two-week break, followed by another cycle of 20 Gy in 2 Gy fx for two weeks. For patients with periampullary cancers, the 2-year OS was 63% in the no adjuvant therapy group and 67% in the CRT group (*p* = 0.737). No difference in local recurrence rates was observed either. Although these data suggest no benefit of CRT, it is important to note that only about 69.2% of patients assigned to CRT actually received the intended treatment, possibly affecting the results [119]. Additionally, it is possible that with modern RT techniques and a greater dose, the results of CRT could be better. Thus, modern prospective data are needed.

Mehta et al. (Table 9) performed a prospective study of 12 patients with “unfavorable” (positive lymph nodes, R1 resection, poor differentiation, a large tumor size, or neurovascular invasion) ampullary carcinoma who received adjuvant 5-FU with concurrent RT (45 Gy) after a pancreaticoduodenectomy. At a median follow-up of 24 months, eight patients remained disease-free. The median actuarial survival was 34 months and the 2-year OS was 89% [120]. A number of retrospective studies on adjuvant RT for ampullary cancers have also been published with mixed results. Some studies showed a benefit of adjuvant RT for all patients or those with high-risk disease features (Table 9) [121,122,123,124]. However, others have found no benefit (Table 9) [125,126,127,128,129,130]. A recently published meta-analysis of 3971 patients receiving adjuvant therapy for ampullary cancer found no difference in the OS when considering all patients (HR 0.998), but showed an improved OS for patients with node involvement (HR 0.627, 95% CI: 0.451–0.87). Adjuvant therapy including RT was found to lead to an improvement in the OS (HR 0.804, 95% CI 0.563–1.149); however, no analysis was conducted for patients with a node-positive disease [131].

As most studies are retrospective and there are no existing modern prospective data, further study is needed. The current recommended approach to adjuvant therapy per the NCCN guidelines depends on the disease stage. For patients with stage I disease, systemic therapy or observation is recommended after surgery. For stage II, adjuvant systemic therapy with or without CRT is recommended, or observation can be recommended. For patients with stage III disease, systemic therapy with or without CRT is recommended. In the adjuvant setting, the RT dose of 45–50.4 Gy in 25–28 fx is generally recommended [118].

### 7.2. Neoadjuvant Radiotherapy

Neoadjuvant RT has also been investigated to downstage the disease prior to potential surgery. In a study by Palta et al. (Table 9), 137 patients with ampullary cancers were included, of which 18 received neoadjuvant CRT and 43 received adjuvant CRT (the median RT dose was 50 Gy). A greater proportion of patients receiving CRT had poorly differentiated disease compared to other included patients. Of the patients receiving neoadjuvant CRT, 28% of the patients achieved a pCR and 67% were downstaged. The three-year local control was better for patients receiving CRT versus no CRT (88% vs. 55%, *p* = 0.001), and was trending towards a benefit in DFS (66% vs. 48%, *p* = 0.09) and OS (62% vs. 46%, *p* = 0.074) [132]. However, no analysis was performed, including only neoadjuvant RT compared to no neoadjuvant treatment. Cloyd et al. (Table 9) performed a retrospective study including 142 patients with ampullary cancer, of which 40 received neoadjuvant CRT or CT and CRT. Of all patients receiving neoadjuvant therapy, 93% of patients received RT (a median RT dose of 50 Gy). Patients who did not receive neoadjuvant therapy were more likely to receive postoperative CT (34.2% versus 14.3%). A total of 14% of patients receiving neoadjuvant therapy demonstrated a pCR. No difference between patients receiving neoadjuvant therapy or no neoadjuvant therapy was observed for locoregional recurrence (7.0% versus 9.1%, *p* > 0.05) or median OS (146 versus 107 months, OR 1.14, 95% CI 0.56–2.31) [133]. Although no benefit of neoadjuvant CRT was observed in this study, the total number of patients receiving neoadjuvant therapy was small and fewer received postoperative CT, possibly skewing the results.

Overall, there are currently limited existing data of neoadjuvant RT for ampullary cancer. Per the NCCN guidelines, neoadjuvant systemic therapy with or without subsequent CRT can be considered for patients with localized disease prior to resection. An RT dose of 45–50.4 Gy in 1.8–2 Gy fx is recommended in the neoadjuvant setting [118].

### 7.3. Definitive/Palliative Radiotherapy

One study was identified investigating non-operative management of ampullary cancer including RT. Facer et al. (Table 9) conducted a study of 2176 patients who did not undergo definitive surgery for ampullary cancer from the NCDB. Most patients did not receive any type of therapy, but 13.1% received palliative CT only, 1.9% received palliative RT, 1.6% received CT and palliative RT, 2.4% received definitive RT, and 9.7% received CT and definitive RT. Definitive RT was most commonly delivered at a dose of 45 Gy or 50.4 Gy. Patients with an increasing disease stage had a tendency to receive CT rather than RT. The one-year OS was 36.7% for those receiving no treatment, 35.1% for those receiving palliative RT only, 53.4% for those receiving palliative CT only, 45.7% for those receiving CT and palliative RT, 56.7% for those receiving definitive RT only, and 59.4% for those receiving definitive RT and CT. The median OS was 7.9, 9.5, 13.1, 10.4, 14.7, and 13.7 months for the same groups, respectively. Patients from the same database undergoing definitive surgery had a one-year OS of 84.3% and median OS of 49.5 months. No significant difference was observed in the one-year OS between the palliative CT group and CT with definitive RT group (HR 0.86, 95% CI 0.72–1.08) [134]. This study suggests no benefit to palliative or definitive RT, but the data quality is poor. The NCCN guidelines currently recommend the consideration of palliative RT for metastatic ampullary cancer, or best supportive care, systemic therapy, or targeted therapy if applicable. In the setting of unresectable, recurrent ampullary cancer, CRT (with an RT dose delivered to 45–56 Gy) can also be considered [117].

**Table 9 curroncol-32-00545-t009:** Selected studies of management of ampullary cancer that include RT.

Study	Type	*n*	Population	Treatment	Key Results
Klinkenbijl et al. (1999) [119]	Phase III Clinical Trial	218; 110 assigned CRT	Patients with T1-2N0-1aM0 cancer of the pancreatic head (55%) or T1-3N0-1aM0 periampullary cancer (45%)	Surgery → CRT (40 Gy in 20 fx with a split course) or observation	For patients with periampullary cancer, no difference in 2-year OS was observed (63% in observation group vs. 67% in treatment group, *p* = 0.737)
Lee et al. (2000) [121]	Single-institution Retrospective	39 total; 13 received RT	Patients with ampullary cancer	Surgery → +/− CRT (median RT dose 48.6 Gy)	When controlling for risk status, adjuvant CRT led to better OS (*p* = 0.03) No benefit was observed for LRFS
Mehta et al. (2001) [120]	Prospective	12	Patients with “unfavorable” ampullary cancer	Surgery → adjuvant CRT (RT to 45 Gy)	Actuarial survival at 2 years: 89% Median OS: 34 months
Sikora et al. (2005) [125]	Single-institution Retrospective	104 total; 49 received RT	Patients with ampullary cancer	Surgery → +/− CRT (median RT dose 50.4 Gy)	There was no significant difference in the median survival (34.6% vs. 24.5%) and 5-year actuarial survival (38% vs. 28%) in the CRT versus no CRT groups, even in high-risk patients
Bhatia et al. (2006) [122]	Single-institution Retrospective	125 total; 29 patients received RT	Patients with ampullary cancer who underwent resection	Surgery → +/− adjuvant CRT (median RT dose 50.4 Gy)	Adjuvant CRT was only beneficial for patients with node-positive disease (median OS 3.4 years vs. 1.6 years, *p* = 0.01)
Krishnan et al. (2008) [126]	Single-institution Retrospective	96 total; 54 received adjuvant RT	Patients with ampullary cancer who underwent resection	Surgery → +/− adjuvant CRT (median RT dose 50.4 Gy)	Adjuvant CRT group demonstrated a trend towards improved OS compared to no CRT group (OS: 35.2 vs. 16.5 months, *p* = 0.06)
Zhou et al. (2009) [127]	Single-institution Retrospective	111 total; 45% received RT	Patients with ampullary cancer who underwent resection	Surgery → +/− adjuvant CRT	On univariate analysis, adjuvant CRT did not improve median OS compared to no adjuvant therapy (33.4 vs. 36.2 months, *p* = 0.969)
Narang et al. (2011) [123]	Multi-institution Retrospective	186 total; 66 received RT	Patients with ampullary cancer who underwent resection	Surgery +/− adjuvant CRT (median total RT dose 50.4 Gy)	When adjusting for negative prognostic factors, adjuvant CRT demonstrated a survival benefit (RR = 0.4, *p* < 0.001)
Palta et al. (2012) [132]	Single-institution Retrospective	137 total; 43 received adjuvant RT and 18 received neoadjuvant RT	Patients with ampullary cancer who underwent resection	+/− Neoadjuvant CRT → Surgery → +/− adjuvant CRT	For patients receiving neoadjuvant CRT, 28% had a pCR 3-year local control was better for patients receiving CRT versus no CRT (88% vs. 55%, *p* = 0.001), and was trending towards benefit in DFS (66% vs. 48%, *p* = 0.09) and OS (62% vs. 46%, *p* = 0.074)
Miura et al. (2014) [128]	Database Retrospective	1287 total; 329 received RT	Patients with ampullary cancer who underwent resection	Surgery → +/− RT	Adjuvant RT did not improve median OS (27 vs. 29 months, *p* = 0.58) or median disease-specific survival (36 vs. 40 months, *p* = 0.92) in propensity-matched cohorts
Kim et al. (2016) [135]	Single-institution Retrospective	71	Patients with ampullary cancer who underwent surgery and RT	Surgery → RT (most received concurrent 5-FU, RT delivered to 40–50 Gy)	5-year LRFS: 76.2% 5-year OS: 64.5%
Pathy et al. (2017) [136]	Single-institution Retrospective	65	Patients with periampullary cancer	Surgery → Adjuvant CRT (RT dose 45 Gy)	Median DFS: 29.64 months One-year OS: 92.7% G3+ hematologic toxicities were experienced by 20%
Cloyd et al. (2017) [133]	Database Retrospective	142 total; 40 received RT	Patients with ampullary cancer	Neoadjuvant CRT, CT, CRT + CT, or no neoadjuvant therapy → surgery	93% of patients receiving neoadjuvant therapy had CRT Receipt of neoadjuvant therapy was not associated with better OS (OR: 1.14, 95% CI: 0.56–2.31)
Kamarajah et al. (2018) [124]	Database Retrospective	1106 total; 27% received RT	Patients with ampullary cancer	Surgery → +/− adjuvant RT	In only patients with N2 disease, RT resulted in improved cancer-specific survival (median 27 vs. 19 months, *p* = 0.0044) and OS (median 23 vs. 17 months, *p* = 0.0091)
Zhao et al. (2019) [129]	Database Retrospective	1227 total; 326 received RT	Patients with resectable ampullary cancer	Surgery → +/− adjuvant RT	RT failed to improve OS or DFS (*p* = 0.119, *p* = 0.188, respectively) In patients older than 70, RT was associated with worse OS and DFS
Manne et al. (2020) [130]	Single-institution Retrospective	63 total; 13 received RT	Patients with resectable ampullary cancer	Surgery → CRT, CT, or no adjuvant therapy	In the CRT vs. CT groups: Median OS was 22.8 vs. 65.7 months (*p* = 0.3975) PFS was 25.3 months vs. 65.7 months (*p* = 0.4699)
Facer et al. (2023) [134]	Database Retrospective	2176 total, 1.9% received palliative RT, 1.6% received CT + palliative RT, 2.4% received definitive RT, and 9.7% received CT + definitive RT	Patients with ampullary cancer who did not undergo resection	Palliative RT CT + palliative RT, definitive RT, or CT + definitive RT	One-year OS ranged from 35.1% (palliative RT) to 59.4% (CT + definitive RT) CT + definitive RT did not lead to improved OS compared to CT only (*p* = 0.87)

## 8. Active Clinical Trials

A search on clinicaltrials.gov reveals 10 active or recruiting studies on RT for CCA. Of note, NCT02773485 is a Phase III randomized trial in India investigating high-dose CRT and CT versus CT (gemcitabine/cisplatin) alone for patients with unresectable CCA. There are active or recruiting studies on RT for GBC. Notably, NCT02867865 is a Phase II/III integrated randomized trial in India investigating preoperative CT (gemcitabine/platinum for four cycles) versus CRT (55–57 Gy with concurrent gemcitabine followed by two cycles of CT) for patients with locally advanced GBC. The interim results were recently presented at ASCO 2025. A greater proportion of patients in the CRT group were able to obtain R0 resection compared to the CT-only group (51.6% versus 29.7%, *p* = 0.01). Using intention-to-treat analysis, the median OS was significantly greater in the CRT group (21.8 months versus 10.1 months, *p* = 0.006) [137]. NCT06214572 is a Phase II/III integrated randomized trial in India investigating systemic therapy (gemcitabine-based) versus systemic and RT (40–55 Gy in 10 fx or up to 60 Gy in 15 fx and gemcitabine-based CT) in patients with unresectable GBC. The final results from these trials will be of great interest.

## 9. Contouring Guidelines

There is limited consensus on the delineation of the CTV for BTCs [138]. Bisello et al. have proposed guidelines for CTV definitions in the definitive setting, including elective nodal irradiation. For IHCC, they recommend a 10 mm radial expansion from the GTV to create the CTV. To delineate the CTV for EHCC, they recommend at least 15 mm radially in all directions, up to 20 mm in the distal direction of the bile duct, and 25 mm in the proximal direction of the bile duct. For GBC, they recommend adding 25 mm radially in the hepatic direction to define the CTV. For regional nodes, in most cases they recommend a 10 mm expansion around the relevant artery/vein/duct. Some exceptions include the lesser gastric curvature lymph nodes, for which the area around the lesser curvature of the stomach should be included; the right pericardial lymph nodes, for which the anatomic space between the gastric cardia and liver, extending posteriorly to the aorta, should be included; and the left paracardial lymph nodes, for which the anatomic space defined medially by the gastric fundus, posteriorly by the spleen, anteromedially by the peritoneum, superiorly by the diaphragm, and inferiorly by the greater curvature lymph nodes should be included [139]. The contouring guidelines for normal organs of the upper GI system have also been proposed [140]. CTV delineation may also be extrapolated from the suggested contouring guidelines for adjuvant RT of pancreatic head cancers [141]. Given the lack of existing guidelines for contouring BTC RT plans, consensus statements from gastrointestinal radiation oncology experts would be of great value, especially in order to increase the role of dose-escalated treatment in the treatment of BTCs.

## 10. Conclusions

Studies investigating RT for BTCs have mostly demonstrated a survival benefit; however, some have also shown no benefit or only a benefit to patients with high-risk disease features. Patients with high-risk disease features, such as positive margins and lymph node involvement, likely gain the greatest benefit from adjuvant RT. Overall, there are a lack of randomized data regarding optimal RT treatment of BTCs, and existing prospective studies tend to have small sample sizes and are not randomized. Randomized data are needed to both determine the benefit of RT and optimal dosing, as multiple studies have demonstrated better disease control with a higher RT dose. Further investigation into treatment modalities such as proton beam therapy should be performed, given that modern techniques may allow for dose escalation and that some studies also demonstrate a high risk for moderate-to-severe gastrointestinal toxicities with RT treatment of biliary tract cancers. Given that surgery is the only known curative treatment option, further investigation of neoadjuvant CRT or RT to improve rates of resection is especially warranted.

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
