# Peer review of "Role of Radiation Therapy for Biliary Tract Cancers"

_curroncol, 2025, doi:10.3390/curroncol32100545_

Round 1

Reviewer 1 Report

Comments and Suggestions for Authors

This review paper summarizes radiotherapy for biliary tract cancer. Biliary tract cancer is a rare type of cancer, and the role of radiotherapy has not been established. In that regard, this review is considered to be of great value.

However, the quality of the paper cannot be said to be high.

First, there is inconsistency in the use of abbreviations. The original terms are repeatedly used before the abbreviations are defined, or the abbreviations are defined but the original terms are used afterward (e.g., EBRT). Furthermore, abbreviations are used without definition (e.g., BED). This is a basic mistake and evidence of low quality, and alone is sufficient grounds for rejection.

Furthermore, the use of terms is inconsistent, with “radiation” and “radiotherapy” used interchangeably, as well as ‘fraction’ and “fracture” being used interchangeably, making the text difficult to read. This should be organized.

Regarding the content, while the sections citing references are acceptable, the authors' personal opinions are intertwined, making it difficult to classify this as a sufficient scientific paper. For example, lines 176 and 177 mention surgery, chemotherapy, and radiotherapy, but there are no citations for the underlying papers, and the authors are merely expressing their personal feelings. Similarly, line 199 mentions several papers on postoperative irradiation, but there are no citations for those either. This renders the paper incomplete as a review article.

There are also other sections that are difficult to classify as scientific papers, and the list is endless.

In summary, this paper should be rejected, and we recommend resubmitting it after improving the overall quality.

Comments on the Quality of English Language

As summarized in the review.
I believe revisions are necessary throughout.
I recommend that the authors reread the entire text.

Reviewer 2 Report

Comments and Suggestions for Authors

Well written summary of evidence on biliary cancer irradiation.

Would suggest adding references to contouring guidelines , with attention to role of elective nodal radiation or not (as in newer contouring guidelines)

also role for dose escalation

Reviewer 3 Report

Comments and Suggestions for Authors

The question of the role of radiation in BTC treatment is a very important topic for BTC oncologists. The authors have successfully reviewed the literature and have done an excellent job summarizing it.

My only concern about the manuscript is the length. As its over 30 pages, it is very comprehensive, but long. I wonder about splitting this into 2 manuscripts, 1 focusing on palliative intent treatment for BTC and one on neoadjuvant and adjuvant rads.

Couple of minor comments:

1. On page 5, first paragraph, I would adjust the sentence "For palliative treatment, chemotherapy has generally been the backbone of management for CCAs" to systemic treatment is the backbone for management .... Doublet chemotherapy by itself is no longer the standard of care for BTC, as immunotherapy is now part of the SOC. As well chemo is not generally the backbone in the palliative intent treatment. it is really the only treatment in advance disease. 

2. Just from an editing point of view, the spacing is not consistent throughout the manuscript and would need to be checked.

3. I am not sure a page and half are needed dedicated to the TMN/AJCC staging. This could be cut down significantly. 

Round 2

Reviewer 3 Report

Comments and Suggestions for Authors

The authors have addressed most of my concerns. I do feel it is still slightly long, but I will leave it to the editors to determine if any changes need to be made. Overall it is a very good manuscript and I have no further comments